# Trimmed Density Ratio Estimation

**Song Liu**[*]
University of Bristol
song.liu@bristol.ac.uk

**Akiko Takeda**
The Institute of Statistical Mathematics,
AIP, RIKEN,
atakeda@ism.ac.jp

**Taiji Suzuki**
University of Tokyo,
Sakigake (PRESTO), JST,
AIP, RIKEN,
taiji@mist.i.u-tokyo.ac.jp

**Kenji Fukumizu**
The Institute of Statistical Mathematics,
fukumizu@ism.ac.jp

## Abstract

Density ratio estimation is a vital tool in both machine learning and statistical community. However, due to the unbounded nature of density ratio, the estimation procedure can be vulnerable to corrupted data points, which often pushes the estimated ratio toward infinity. In this paper, we present a robust estimator which automatically identifies and trims outliers. The proposed estimator has a convex formulation, and the global optimum can be obtained via subgradient descent. We analyze the parameter estimation error of this estimator under high-dimensional settings. Experiments are conducted to verify the effectiveness of the estimator.

## 1   Introduction

Density ratio estimation (DRE) [18, 11, 27] is an important tool in various branches of machine learning and statistics. Due to its ability of directly modelling the differences between two probability density functions, DRE finds its applications in change detection [13, 6], two-sample test [32] and outlier detection [1, 26]. In recent years, a sampling framework called Generative Adversarial Network (GAN) (see e.g., [9, 19]) uses the density ratio function to compare artificial samples from a generative distribution and real samples from an unknown distribution. DRE has also been widely discussed in statistical literatures for adjusting non-parametric density estimation [5], stabilizing the estimation of heavy tailed distribution [7] and fitting multiple distributions at once [8].

However, as a density ratio function can grow unbounded, DRE can suffer from robustness and stability issues: a few corrupted points may completely mislead the estimator (see Figure 2 in Section 6 for example). Considering a density ratio $p(x)/q(x)$, a point $x$ that is extremely far away from the high density region of $q$ may have an almost infinite ratio value and DRE results can be *dominated* by such points. This makes DRE performance very sensitive to rare pathological data or small modifications of the dataset. Here we give two examples:

**Cyber-attack**   In change detection applications, a density ratio $p(x)/q(x)$ is used to determine how the data generating model differs between $p$ and $q$. Consider a "hacker" who can spy on our data may just inject a few data points in $p$ which are extremely far away from the high-density region of $q$. This would result excessively large $p(x)/q(x)$ tricking us to believe there is a significant change from $q(x)$ to $p(x)$, even if there is no change at all. If the generated outliers are also far away from the

---

[*]This work was done when Song Liu was at The Institute of Statistical Mathematics, Japan

high density region of $p(x)$, we end up with a very different density ratio function and the original parametric pattern in the ratio is ruined. We give such an example in Section 6.

**Volatile Samples**   The change of external environment may be responded in unpredictable ways. It is possible that a small portion of samples react more "aggressively" to the change than the others. These samples may be skewed and show very high density ratios, even if the change of distribution is relatively mild when these volatile samples are excluded. For example, when testing a new fertilizer, a small number of plants may fail to adapt, even if the vast majority of crops are healthy.

Overly large density ratio values can cause further troubles when the ratio is used to weight samples. For example, in the domain adaptation setting, we may reweight samples from one task and reuse them in another task. Density ratio is a natural choice of such "importance weighting" scheme [28, 25]. However, if one or a few samples have extremely high ratio, after renormalizing, other samples will have almost zero weights and have little impact to the learning task.

Several methods have been proposed to solve this problem. The relative density ratio estimation [33] estimates a "biased" version of density ratio controlled by a mixture parameter $\alpha$. The relative density ratio is always upper-bounded by $\frac{1}{\alpha}$, which can give a more robust estimator. However, it is not clear how to de-bias such an estimator to recover the true density ratio function. [26] took a more direct approach. It estimates a *thresholded* density ratio by setting up a tolerance $t$ to the density ratio value. All likelihood ratio values bigger than $t$ will be clipped to $t$. The estimator was derived from Fenchel duality for $f$-divergence [18]. However, the optimization for the estimator is not convex if one uses log-linear models. The formulation also relies on the non-parametric approximation of the density ratio function (or the log ratio function) making the learned model hard to interpret. Moreover, there is no intuitive way to directly control the proportion of ratios that are thresholded. Nonetheless, the concept studied in our paper is inspired by this pioneering work.

In this paper, we propose a novel method based on a "trimmed Maximum Likelihood Estimator" [17, 10]. This idea relies on a specific type of density ratio estimator (called log-linear KLIEP) [30] which can be written as a maximum likelihood formulation. We simply "ignore" samples that make the empirical likelihood take exceedingly large values. The trimmed density ratio estimator can be formulated as a convex optimization and translated into a weighted M-estimator. This helps us develop a simple subgradient-based algorithm that is guaranteed to reach the global optimum.

Moreover, we shall prove that in addition to recovering the correct density ratio under the outlier setting, the estimator can also obtain a "corrected" density ratio function under a truncation setting. It ignores "pathological" samples and recovers density ratio only using "healthy" samples.

Although trimming will usually result a more robust estimate of the density ratio function, we also point out that it should not be abused. For example, in the tasks of two-sample test, a diverging density ratio might indicate interesting structural differences between two distributions.

In Section 2, we explain some preliminaries on trimmed maximum likelihood estimator. In Section 3, we introduce a trimmed DRE. We solve it using a convex formulation whose optimization procedure is explained in Section 4. In Section 5, we prove the estimation error upper-bound with respect to a sparsity inducing regularizer. Finally, experimental results are shown in Section 6 and we conclude our work in Section 7.

## 2   Preliminary: Trimmed Maximum Likelihood Estimation

Although our main purpose is to estimate the density ratio, we first introduce the basic concept of *trimmed estimator* using density functions as examples. Given $n$ samples drawn from a distribution $P$, i.e., $X := \left\{\boldsymbol{x}^{(i)}\right\}_{i=1}^{n} \overset{\text{i.i.d.}}{\sim} P, \boldsymbol{x} \in \mathbb{R}^d$, we want to estimate the density function $p(\boldsymbol{x})$. Suppose the true density function is a member of *exponential family* [20],

$$p(\boldsymbol{x}; \boldsymbol{\theta}) = \exp\left[\langle \boldsymbol{\theta}, \boldsymbol{f}(\boldsymbol{x}) \rangle - \log Z(\boldsymbol{\theta})\right], \;\; Z(\boldsymbol{\theta}) = \int q(\boldsymbol{x}) \exp\langle \boldsymbol{\theta}, \boldsymbol{f}(\boldsymbol{x}) \rangle d\boldsymbol{x} \tag{1}$$

where $\boldsymbol{f}(\boldsymbol{x})$ is the sufficient statistics, $Z(\boldsymbol{\theta})$ is the normalization function and $q(\boldsymbol{x})$ is the base measure.

Maximum Likelihood Estimator (MLE) maximizes the empirical likelihood over the entire dataset. In contrast, a *trimmed* MLE only maximizes the likelihood over a *subset* of samples according to

their likelihood values (see e.g., [10, 31]). This paradigm can be used to derive a popular outlier detection method, one-class Support Vector Machine (one-SVM) [24]. The derivation is crucial to the development of our trimmed density ratio estimator in later sections.

Without loss of generality, we can set the log likelihood function as $\log p(\boldsymbol{x}^{(i)};\boldsymbol{\theta}) - \tau_0$, where $\tau_0$ is a constant. As samples corresponding to high likelihood values are likely to be inliers, we can trim all samples whose likelihood is bigger than $\tau_0$ using a clipping function $[\cdot]_-$, i.e., $\hat{\boldsymbol{\theta}} = \arg\max_{\boldsymbol{\theta}} \sum_{i=1}^{n}[\log p(\boldsymbol{x}^{(i)};\boldsymbol{\theta}) - \tau_0]_-$, where $[\ell]_-$ returns $\ell$ if $\ell \leq 0$ and 0 otherwise. This optimization has a *convex* formulation:

$$\min_{\boldsymbol{\theta},\boldsymbol{\epsilon}\geq 0} \langle\boldsymbol{\epsilon}, \mathbf{1}\rangle, \quad \text{s.t. } \forall i, \log p\left(\boldsymbol{x}^{(i)};\boldsymbol{\theta}\right) \geq \tau_0 - \epsilon_i, \tag{2}$$

where $\boldsymbol{\epsilon}$ is the slack variable measuring the difference between $\log p\left(\boldsymbol{x}^{(i)};\boldsymbol{\theta}\right)$ and $\tau_0$. However, formulation (2) is not practical since computing the normalization term $Z(\boldsymbol{\theta})$ in (1) is intractable for a general $\boldsymbol{f}$ and it is unclear how to set the trimming level $\tau_0$. Therefore we ignore the normalization term and introduce other control terms:

$$\min_{\boldsymbol{\theta},\boldsymbol{\epsilon}\geq 0,\tau\geq 0} \frac{1}{2}\|\boldsymbol{\theta}\|^2 - \nu\tau + \frac{1}{n}\langle\boldsymbol{\epsilon}, \mathbf{1}\rangle \quad \text{s.t. } \forall i, \langle\boldsymbol{\theta}, \boldsymbol{f}(\boldsymbol{x}^{(i)})\rangle \geq \tau - \epsilon_i. \tag{3}$$

The $\ell_2$ regularization term is introduced to avoid $\boldsymbol{\theta}$ reaching unbounded values. A new hyper parameter $\nu \in (0, 1]$ replaces $\tau_0$ to control the number of trimmed samples. It can be proven using KKT conditions that at most $1 - \nu$ fraction of samples are discarded (see e.g., [24], Proposition 1 for details). Now we have reached the standard formulation of one-SVM.

This trimmed estimator ignores the large likelihood values and creates a focus only on the low density region. Such a trimming strategy allows us to discover "novel" points or outliers which are usually far away from the high density area.

## 3  Trimmed Density Ratio Estimation

In this paper, our main focus is to derive a *robust* density ratio estimator following a similar trimming strategy. First, we briefly review the a density ratio estimator [27] from the perspective of Kullback-Leibler divergence minimization.

### 3.1  Density Ratio Estimation (DRE)

For two sets of data $X_p := \{\boldsymbol{x}_p^{(1)}, \ldots, \boldsymbol{x}_p^{(n_p)}\} \overset{\text{i.i.d.}}{\sim} P$, $X_q := \{\boldsymbol{x}_q^{(1)}, \ldots, \boldsymbol{x}_q^{(n_q)}\} \overset{\text{i.i.d.}}{\sim} Q$, assume both the densities $p(\boldsymbol{x})$ and $q(\boldsymbol{x})$ are in exponential family (1). We know $\frac{p(\boldsymbol{x};\boldsymbol{\theta}_p)}{q(\boldsymbol{x};\boldsymbol{\theta}_q)} \propto \exp\left[\langle\boldsymbol{\theta}_p - \boldsymbol{\theta}_q, \boldsymbol{f}(\boldsymbol{x})\rangle\right]$. Observing that the data $\boldsymbol{x}$ only interacts with the parameter $\boldsymbol{\theta}_p - \boldsymbol{\theta}_q$ through $\boldsymbol{f}$, we can keep using $\boldsymbol{f}(\boldsymbol{x})$ as our sufficient statistic for the density ratio model, and merge two parameters $\boldsymbol{\theta}_p$ and $\boldsymbol{\theta}_q$ into one single parameter $\boldsymbol{\delta} = \boldsymbol{\theta}_p - \boldsymbol{\theta}_q$. Now we can model our density ratio as

$$r(\boldsymbol{x};\boldsymbol{\delta}) := \exp\left[\langle\boldsymbol{\delta}, \boldsymbol{f}(\boldsymbol{x})\rangle - \log N(\boldsymbol{\delta})\right], \ N(\boldsymbol{\delta}) := \int q(\boldsymbol{x})\exp\langle\boldsymbol{\delta}, \boldsymbol{f}(\boldsymbol{x})\rangle d\boldsymbol{x}, \tag{4}$$

where $N(\boldsymbol{\delta})$ is the normalization term that guarantees $\int q(\boldsymbol{x})r(\boldsymbol{x};\boldsymbol{\delta})d\boldsymbol{x} = 1$ so that $q(\boldsymbol{x})r(\boldsymbol{x};\boldsymbol{\delta})$ is a valid density function and is normalized over its domain.

Interestingly, despite the parameterization (changing from $\boldsymbol{\theta}$ to $\boldsymbol{\delta}$), (4) is exactly the same as (1) where $q(\boldsymbol{x})$ appeared as a base measure. The difference is, here, $q(\boldsymbol{x})$ is a *density function* from which $X_q$ are drawn so that $N(\boldsymbol{\delta})$ can be approximated accurately from samples of $Q$. Let us define

$$\hat{r}(\boldsymbol{x};\boldsymbol{\delta}) := \exp\left[\langle\boldsymbol{\delta}, \boldsymbol{f}(\boldsymbol{x})\rangle - \log \widehat{N}(\boldsymbol{\delta})\right], \ \widehat{N}(\boldsymbol{\delta}) := \frac{1}{n_q}\sum_{j=1}^{n_q}\exp\left[\langle\boldsymbol{\delta}, \boldsymbol{f}(\boldsymbol{x}_q^{(j)})\rangle\right]. \tag{5}$$

Note this model can be computed for any $\boldsymbol{f}$ even if the integral in $N(\boldsymbol{\delta})$ does not have a closed form .

In order to estimate $\boldsymbol{\delta}$, we minimize the Kullback-Leibler divergence between $p$ and $q \cdot r_{\boldsymbol{\delta}}$:

$$\min_{\boldsymbol{\delta}} \mathrm{KL}\left[p | q \cdot r_{\boldsymbol{\delta}}\right] = \min_{\boldsymbol{\delta}} \int p(\boldsymbol{x}) \log \frac{p(\boldsymbol{x})}{q(\boldsymbol{x}) r(\boldsymbol{x}; \boldsymbol{\delta})} d\boldsymbol{x} = c - \max_{\boldsymbol{\delta}} \int p(\boldsymbol{x}) \log r(\boldsymbol{x}; \boldsymbol{\delta}) d\boldsymbol{x}$$

$$\approx c - \max_{\boldsymbol{\delta}} \frac{1}{n_p} \sum_{i=1}^{n_p} \log \hat{r}(\boldsymbol{x}_p^{(i)}; \boldsymbol{\delta}) \tag{6}$$

where $c$ is a constant irrelevant to $\boldsymbol{\delta}$. It can be seen that the minimization of KL divergence boils down to *maximizing log likelihood ratio* over dataset $X_p$.

Now we have reached the log-linear Kullback-Leibler Importance Estimation Procedure (log-linear KLIEP) estimator [30, 14].

## 3.2 Trimmed Maximum Likelihood Ratio

As stated in Section 1, to rule out the influences of large density ratio, we trim samples with large likelihood ratio values from (6). Similarly to one-SVM in (2), we can consider a trimmed MLE $\hat{\boldsymbol{\delta}} = \arg\max_{\boldsymbol{\delta}} \sum_{i=1}^{n_p} [\log \hat{r}(\boldsymbol{x}_p^{(i)}; \boldsymbol{\delta}) - t_0]_-$ where $t_0$ is a threshold above which the likelihood ratios are ignored. It has a convex formulation:

$$\min_{\boldsymbol{\delta}, \boldsymbol{\epsilon} \geq \mathbf{0}} \langle \boldsymbol{\epsilon}, \mathbf{1} \rangle, \quad \text{s.t. } \forall \boldsymbol{x}_p^{(i)} \in X_p, \log \hat{r}(\boldsymbol{x}_p^{(i)}; \boldsymbol{\delta}) \geq t_0 - \epsilon_i. \tag{7}$$

(7) is similar to (2) since we have only replaced $p(\boldsymbol{x}; \boldsymbol{\theta})$ with $\hat{r}(\boldsymbol{x}; \boldsymbol{\delta})$. However, the ratio model $\hat{r}(\boldsymbol{x}; \boldsymbol{\delta})$ in (7) comes with a tractable normalization term $\hat{N}$ while the normalization term $Z$ in $p(\boldsymbol{x}; \boldsymbol{\theta})$ is in general intractable.

Similar to (3), we can directly control the trimming quantile via a hyper-parameter $\nu$:

$$\min_{\boldsymbol{\delta}, \boldsymbol{\epsilon} \geq \mathbf{0}, t \geq 0} \frac{1}{n_p} \langle \boldsymbol{\epsilon}, \mathbf{1} \rangle - \nu \cdot t + \lambda R(\boldsymbol{\delta}), \quad \text{s.t. } \forall \boldsymbol{x}_p^{(i)} \in X_p, \log \hat{r}(\boldsymbol{x}_p^{(i)}; \boldsymbol{\delta}) \geq t - \epsilon_i \tag{8}$$

where $R(\boldsymbol{\delta})$ is a convex regularizer. (8) is also convex, but it has $n_p$ number of *non-linear* constraints and the search for the global optimal solution can be time-consuming. To avoid such a problem, one could derive and solve the dual problem of (8). In some applications, we rely on the primal parameter structure (such as sparsity) for model interpretation, and feature engineering. In Section 4, we translate (8) into an equivalent form so that its solution is obtained via a subgradient ascent method which is guaranteed to converge to the global optimum.

One common way to construct a convex robust estimator is using a Huber loss [12]. Although the proposed trimming technique rises from a different setting, it shares the same guiding principle with Huber loss: avoid assigning dominating values to outlier likelihoods in the objective function.

In Section 8.1 in the supplementary material, we show the relationship between trimmed DRE and binary Support Vector Machines [23, 4].

# 4 Optimization

The key to solving (8) efficiently is reformulating it into an equivalent $\max\min$ problem.

**Proposition 1.** *Assuming $\nu$ is chosen such that $\hat{t} > 0$ for all optimal solutions in (8), then $\hat{\boldsymbol{\delta}}$ is an optimal solution of (8) if and only if it is also the optimal solution of the following $\max\min$ problem:*

$$\max_{\boldsymbol{\delta}} \min_{\boldsymbol{w} \in \left[0, \frac{1}{n_p}\right]^{n_p}, \langle \mathbf{1}, \boldsymbol{w} \rangle = \nu} \mathcal{L}(\boldsymbol{\delta}, \boldsymbol{w}) - \lambda R(\boldsymbol{\delta}), \ \mathcal{L}(\boldsymbol{\delta}, \boldsymbol{w}) := \sum_{i=1}^{n_p} w_i \cdot \log \hat{r}(\boldsymbol{x}_p^{(i)}; \boldsymbol{\delta}). \tag{9}$$

The proof is in Section 8.2 in the supplementary material. We define $(\hat{\boldsymbol{\delta}}, \hat{\boldsymbol{w}})$ as a saddle point of (9):

$$\nabla_{\boldsymbol{\delta}} \mathcal{L}(\hat{\boldsymbol{\delta}}, \hat{\boldsymbol{w}}) - \nabla_{\boldsymbol{\delta}} \lambda R(\hat{\boldsymbol{\delta}}) = \mathbf{0}, \hat{\boldsymbol{w}} \in \arg \min_{\boldsymbol{w} \in [0, \frac{1}{n_p}]^{n_p}, \langle \boldsymbol{w}, \mathbf{1} \rangle = \nu} \mathcal{L}(\hat{\boldsymbol{\delta}}, \boldsymbol{w}), \tag{10}$$

where the second $\nabla_{\boldsymbol{\delta}}$ means the subgradient if $R$ is sub-differentiable.

---
**Algorithm 1** Gradient Ascent and Trimming
---
Input: $X_p, X_q, \nu$ and step sizes $\{\eta_{\mathrm{it}}\}_{\mathrm{it}=1}^{\mathrm{it_{max}}}$; Initialize $\boldsymbol{\delta}_0, \boldsymbol{w}_0$, Iteration counter: it $= 0$, Maximum number of iterations: $\mathrm{it_{max}}$, Best objective, parameter pair $(O_{\mathrm{best}} = -\infty, \boldsymbol{\delta}_{\mathrm{best}}, \boldsymbol{w}_{\mathrm{best}})$ .

**while** not converged and it $\leq \mathrm{it_{max}}$ **do**

    Obtain a sorted set $\left\{\boldsymbol{x}_p^{(i)}\right\}_{i=1}^{n_p}$ so that $\log \hat{r}(\boldsymbol{x}_p^{(1)}; \boldsymbol{\delta}_{\mathrm{it}}) \leq \log \hat{r}(\boldsymbol{x}_p^{(2)}; \boldsymbol{\delta}_{\mathrm{it}}) \cdots \leq \log \hat{r}(\boldsymbol{x}_p^{(n_p)}; \boldsymbol{\delta}_{\mathrm{it}})$.

    $w_{\mathrm{it}+1,i} = \frac{1}{n_p}, \forall i \leq \nu n_p$. $w_{\mathrm{it}+1,i} = 0$, otherwise.

    Gradient ascent with respect to $\boldsymbol{\delta}$: $\boldsymbol{\delta}_{\mathrm{it}+1} = \boldsymbol{\delta}_{\mathrm{it}} + \eta_{\mathrm{it}} \cdot \nabla_{\boldsymbol{\delta}}[\mathcal{L}(\boldsymbol{\delta}_{\mathrm{it}}, \boldsymbol{w}_{\mathrm{it}+1}) - \lambda R(\boldsymbol{\delta}_{\mathrm{it}})]$,

    $O_{\mathrm{best}} = \max(O_{\mathrm{best}}, \mathcal{L}(\boldsymbol{\delta}_{\mathrm{it}+1}, \boldsymbol{w}_{\mathrm{it}+1}))$ and update $(\boldsymbol{\delta}_{\mathrm{best}}, \boldsymbol{w}_{\mathrm{best}})$ accordingly.   it $=$ it $+ 1$.

**end while**

Output: $(\boldsymbol{\delta}_{\mathrm{best}}, \boldsymbol{w}_{\mathrm{best}})$
---

Now the "trimming" process of our estimator can be clearly seen from (9): The $\max$ procedure estimates a density ratio given the currently assigned weights $\boldsymbol{w}$, and the $\min$ procedure trims the large log likelihood ratio values by assigning corresponding $w_i$ to 0 (or values smaller than $\frac{1}{n_p}$). For simplicity, we only consider the cases where $\nu$ is a multiple of $\frac{1}{n_p}$. Intuitively, $1 - \nu$ is the proportion of likelihood ratios that are trimmed thus $\nu$ should not be greater than 1. Note if we set $\nu = 1$, (9) is equivalent to the standard density ratio estimator (6). Downweighting outliers while estimating the model parameter $\boldsymbol{\delta}$ is commonly used by robust estimators (See e.g., [3, 29]).

The search for $(\hat{\boldsymbol{\delta}}, \hat{\boldsymbol{w}})$ is straightforward. It is easy to solve with respect to $\boldsymbol{w}$ or $\boldsymbol{\delta}$ while the other is fixed: given a parameter $\boldsymbol{\delta}$, the optimization with respect to $\boldsymbol{w}$ is a linear programming and *one of* the extreme optimal solutions is attained by assigning weight $\frac{1}{n_p}$ to the elements that correspond to the $\nu n_p$-smallest log-likelihood ratio $\log \hat{r}(\boldsymbol{x}^{(i)}, \boldsymbol{\delta})$. This observation leads to a simple "gradient ascent and trimming" algorithm (see Algorithm 1). In Algorithm 1,

$$\nabla_{\boldsymbol{\delta}} \mathcal{L}(\boldsymbol{\delta}, \boldsymbol{w}) = \frac{1}{n_p} \sum_{i=1}^{n_p} w_i \cdot \boldsymbol{f}(\boldsymbol{x}_p^{(i)}) - \nu \cdot \sum_{j=1}^{n_q} \frac{e^{(j)}}{\sum_{k=1}^{n_q} e^{(k)}} \boldsymbol{f}(x_q^{(j)}), \;\; e^{(i)} := \exp(\langle \boldsymbol{\delta}, \boldsymbol{f}(x_q^{(i)}) \rangle).$$

In fact, Algorithm 1 is a subgradient method [2, 16], since the optimal value function of the inner problem of (9) is not differentiable at some $\boldsymbol{\delta}$ where the inner problem has multiple optimal solutions. The subdifferential of the optimal value of the inner problem with respect to $\boldsymbol{\delta}$ can be a *set* but Algorithm 1 only computes a subgradient obtained using the extreme point solution $\boldsymbol{w}_{\mathrm{it}+1}$ of the inner linear programming. Under mild conditions, this subgradient ascent approach will converge to optimal results with diminishing step size rule and it $\to \infty$. See [2] for details.

Algorithm 1 is a simple gradient ascent procedure and can be implemented by deep learning softwares such as Tensorflow[2] which benefits from the GPU acceleration. In contrast, the original problem (8), due to its heavily constrained nature, cannot be easily programmed using such a framework.

## 5   Estimation Consistency in High-dimensional Settings

In this section, we show how the estimated parameter $\hat{\boldsymbol{\delta}}$ in (10) converges to the "optimal parameters" $\boldsymbol{\delta}^*$ as both sample size and dimensionality goes to infinity under the "outlier" and "truncation" setting respectively.

In the **outlier setting** (Figure 1a), we assume $X_p$ is contaminated by outliers and all "inlier" samples in $X_p$ are i.i.d.. The outliers are injected into our dataset $X_p$ after looking at our inliers. For example, hackers can spy on our data and inject fake samples so that our estimator exaggerates the degree of change.

In the **truncation setting**, there are no outliers. $X_p$ and $X_q$ are i.i.d. samples from $P$ and $Q$ respectively. However, we have a subset of "volatile" samples in $X_p$ (the rightmost mode on histogram in Figure 1b) that are pathological and exhibit large density ratio values.

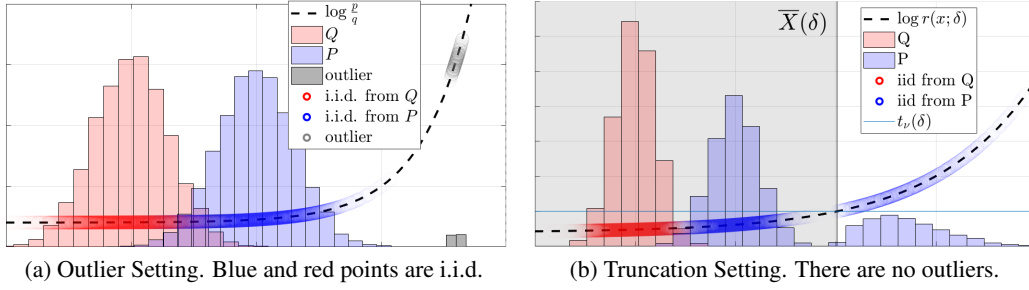

(a) Outlier Setting. Blue and red points are i.i.d.   (b) Truncation Setting. There are no outliers.

Figure 1: Two settings of theoretical analysis.

In the theoretical results in this section, we focus on analyzing the performance of our estimator for high-dimensional data assuming the number of non-zero elements in the optimal $\boldsymbol{\delta}^*$ is $k$ and use the $\ell_1$ regularizer, i.e., $R(\boldsymbol{\theta}) = \|\boldsymbol{\theta}\|_1$ which induces sparsity on $\hat{\boldsymbol{\delta}}$. The proofs rely on a recent development [35, 34] where a "weighted" high-dimensional estimator was studied. We also assume the optimization of $\boldsymbol{\delta}$ in (9) was conducted within an $\ell_1$ ball of width $\rho$, i.e., $\mathrm{Ball}(\rho)$, and $\rho$ is wisely chosen so that the optimal parameter $\boldsymbol{\delta}^* \in \mathrm{Ball}(\rho)$. The same technique was used in previous works [15, 35].

**Notations:**   We denote $\boldsymbol{w}^* \in \mathbb{R}^{n_p}$ as the "optimal" weights depending on $\boldsymbol{\delta}^*$ and our data. To lighten the notation, we shorten the *log* density ratio model as $z_{\boldsymbol{\delta}}(\boldsymbol{x}) := \log r(\boldsymbol{x}; \boldsymbol{\delta})$, $\hat{z}_{\boldsymbol{\delta}}(\boldsymbol{x}) := \log \hat{r}(\boldsymbol{x}; \boldsymbol{\delta})$

The proof of Theorem 1, 2 and 3 can be found in Section 8.4, 8.5 and 8.6 in supplementary materials.

## 5.1   A Base Theorem

Now we provide a base theorem giving an upperbound of $\|\hat{\boldsymbol{\delta}} - \boldsymbol{\delta}^*\|$. We state this theorem only with respect to an arbitrary pair $(\boldsymbol{\delta}^*, \boldsymbol{w}^*)$ and the pair is set properly later in Section 5.2 and 5.3.

We make a few regularity conditions on samples from $Q$. Samples of $X_q$ should be well behaved in terms of log-likelihood ratio values.

**Assumption 1.** $\exists 0 < c_1 < 1, 1 < c_2 < \infty \; \forall \boldsymbol{x}_q \in X_q, \boldsymbol{u} \in \mathrm{Ball}(\rho), c_1 \leq \exp\langle \boldsymbol{\delta}^* + \boldsymbol{u}, \boldsymbol{x}_q \rangle \leq c_2$ *and collectively* $c_2/c_1 = C_r$.

We also assume the Restricted Strong Convexity (RSC) condition on the covariance of $\boldsymbol{X}_q$, i.e., $\mathrm{cov}(\boldsymbol{X}_q) = \frac{1}{n_q}(\boldsymbol{X}_q - \frac{1}{n_q}\boldsymbol{X}_q\boldsymbol{1})(\boldsymbol{X}_q - \frac{1}{n_q}\boldsymbol{X}_q\boldsymbol{1})^\top$. Note this property has been verified for various different design matrices $\boldsymbol{X}_q$, such as Gaussian or sub-Gaussian (See, e.g., [21, 22]).

**Assumption 2.** *RSC condition of* $\mathrm{cov}(\boldsymbol{X}_q)$ *holds for all* $\boldsymbol{u}$, *i.e., there exists* $\kappa_1' > 0$ *and* $c > 0$ *such that* $\boldsymbol{u}^\top \mathrm{cov}(\boldsymbol{X}_q)\boldsymbol{u} \geq \kappa_1'\|\boldsymbol{u}\|^2 - \frac{c}{\sqrt{n_q}}\|\boldsymbol{u}\|_1^2$ *with high probability.*

**Theorem 1.** *In addition to Assumption 1 and 2, there exists coherence between parameter* $\boldsymbol{w}$ *and* $\boldsymbol{\delta}$ *at a saddle point* $(\hat{\boldsymbol{\delta}}, \hat{\boldsymbol{w}})$:

$$\langle \nabla_{\boldsymbol{\delta}}\mathcal{L}(\hat{\boldsymbol{\delta}}, \hat{\boldsymbol{w}}) - \nabla_{\boldsymbol{\delta}}\mathcal{L}(\hat{\boldsymbol{\delta}}, \boldsymbol{w}^*), \hat{\boldsymbol{u}} \rangle \geq -\kappa_2\|\hat{\boldsymbol{u}}\|^2 - \tau_2(n,d)\|\hat{\boldsymbol{u}}\|_1, \tag{11}$$

*where* $\hat{\boldsymbol{u}} := \hat{\boldsymbol{\delta}} - \boldsymbol{\delta}^*$, $\kappa_2 > 0$ *is a constant and* $\tau_2(d, n) > 0$. *It can be shown that if*

$$\lambda_n \geq 2\max\left[\|\nabla_{\boldsymbol{\delta}}\mathcal{L}(\boldsymbol{\delta}^*, \boldsymbol{w}^*)\|_\infty, \frac{\rho\nu c}{2C_r^2\sqrt{n_q}}, \tau_2(n,d)\right]$$

*and* $\nu\kappa_1' > 2C_r^2\kappa_2$, *where* $c > 0$ *is a constant determined by RSC condition, we are guaranteed that* $\|\hat{\boldsymbol{\delta}} - \boldsymbol{\delta}^*\| \leq \frac{C_r^2}{(\nu\kappa_1' - 2C_r^2\kappa_2)} \cdot \frac{3\sqrt{k}\lambda_n}{2}$ *with probability converging to one.*

The condition (11) states that if we swap $\hat{\boldsymbol{w}}$ for $\boldsymbol{w}^*$, the change of the gradient $\nabla_{\boldsymbol{\delta}}\mathcal{L}$ is limited. Intuitively, it shows that our estimator (9) is not "picky" on $\boldsymbol{w}$: even if we cannot have the optimal weight assignment $\boldsymbol{w}^*$, we can still use "the next best thing", $\hat{\boldsymbol{w}}$ to compute the gradient which is close enough. We later show how (11) is satisfied. Note if $\|\nabla_{\boldsymbol{\delta}}\mathcal{L}(\boldsymbol{\delta}^*, \boldsymbol{w}^*)\|_\infty, \tau_2(n,d)$ converge to zero as $n_p, n_q, d \to \infty$, by taking $\lambda_n$ as such, Theorem 1 guarantees the consistency of $\hat{\boldsymbol{\delta}}$. In Section 5.2 and 5.3, we explore two different settings of $(\boldsymbol{\delta}^*, \boldsymbol{w}^*)$ that make $\|\hat{\boldsymbol{\delta}} - \boldsymbol{\delta}^*\|$ converges to zero.

## 5.2 Consistency under Outlier Setting

**Setting:** Suppose dataset $X_p$ is the union of two disjoint sets $G$ (Good points) and $B$ (Bad points) such that $G \overset{\text{i.i.d.}}{\sim} p(\boldsymbol{x})$ and $\min_{j \in B} z_{\boldsymbol{\delta}^*}(\boldsymbol{x}_p^{(j)}) > \max_{i \in G} z_{\boldsymbol{\delta}^*}(\boldsymbol{x}_p^{(i)})$ (see Figure 1a). Dataset $X_q \overset{\text{i.i.d.}}{\sim} q(\boldsymbol{x})$ does *not* contain any outlier. We set $\nu = \frac{|G|}{n_p}$. The optimal parameter $\boldsymbol{\delta}^*$ is set such that $p(\boldsymbol{x}) = q(\boldsymbol{x})r(\boldsymbol{x}; \boldsymbol{\delta}^*)$. We set $w_i^* = \frac{1}{n_p}, \forall \boldsymbol{x}_p^{(i)} \in G$ and 0 otherwise.

**Remark:** Knowing the inlier proportion $|G|/n_p$ is a strong assumption. However it is only imposed for theoretical analysis. As we show in Section 6, our method works well even if $\nu$ is only a rough guess (like $90\%$). Loosening this assumption will be an important future work.

**Assumption 3.** $\forall \boldsymbol{u} \in \mathrm{Ball}(\rho), \sup_{\boldsymbol{x}} |\hat{z}_{\boldsymbol{\delta}^* + \boldsymbol{u}}(\boldsymbol{x}) - \hat{z}_{\boldsymbol{\delta}^*}(\boldsymbol{x})| \leq C_{\mathrm{lip}}\|\boldsymbol{u}\|_1$.

This assumption says that the log density ratio model is Lipschitz continuous around its optimal parameter $\boldsymbol{\delta}^*$ and hence there is a limit how much a log ratio model can deviate from the optimal model under a small perturbation $\boldsymbol{u}$. As our estimated weights $\hat{w}_i$ depends on the relative ranking of $\hat{z}_{\hat{\boldsymbol{\delta}}}(\boldsymbol{x}_p^{(i)})$, this assumption implies that the relative ranking between two points will remain unchanged under a small perturbation $\boldsymbol{u}$ if they are far apart. The following theorem shows that if we have enough clearance between "good" and "bad samples", $\hat{\boldsymbol{\delta}}$ converges to the optimal parameter $\boldsymbol{\delta}^*$.

**Theorem 2.** *In addition to Assumption 1, 2 and a few mild technical conditions (see Section 8.5 in the supplementary material), Assumptions 3 holds. Suppose $\min_{j \in B} z_{\boldsymbol{\delta}^*}(\boldsymbol{x}_p^{(j)}) - \max_{i \in G} z_{\boldsymbol{\delta}^*}(\boldsymbol{x}_p^{(i)}) \geq 3C_{\mathrm{lip}}\rho, \nu = \frac{|G|}{n_p}, n_q = \Omega(|G|^2)$. If $\lambda_n \geq 2 \cdot \max \left( \sqrt{\frac{K_1 \log d}{|G|}}, \frac{\rho\nu c}{2C_r^2 \sqrt{n_q}} \right)$, where $K_1 > 0, c > 0$ are constants, we are guaranteed that $\|\hat{\boldsymbol{\delta}} - \boldsymbol{\delta}^*\| \leq \frac{C_r^2}{\nu \kappa_1'} \cdot 3\sqrt{k}\lambda_n$ with probability converging to 1.*

It can be seen that $\|\hat{\boldsymbol{\delta}} - \boldsymbol{\delta}^*\| = O\left( \sqrt{\log d / \min(|G|, n_q)} \right)$ if $d$ is reasonably large.

## 5.3 Consistency under Truncation Setting

In this setting, we do not assume there are outliers in the observed data. Instead, we examine the ability of our estimator recovering the density ratio up to a certain quantile of our data. This ability is especially useful when the behavior of the tail quantile is more volatile and makes the standard estimator (6) output unpredictable results.

**Notations:** Given $\nu \in (0, 1]$, we call $t_\nu(\boldsymbol{\delta})$ is the $\nu$-th quantile of $z_{\boldsymbol{\delta}}$ if $P\left[z_{\boldsymbol{\delta}} < t_\nu(\boldsymbol{\delta})\right] \leq \nu$ and $P\left[z_{\boldsymbol{\delta}} \leq t_\nu(\boldsymbol{\delta})\right] \geq \nu$. In this setting, we consider $\nu$ is fixed by a user thus we drop the subscript $\nu$ from all subsequent discussions. Let's define a truncated domain: $\overline{X}(\boldsymbol{\delta}) = \left\{ \boldsymbol{x} \in \mathbb{R}^d | z_{\boldsymbol{\delta}}(\boldsymbol{x}) < t(\boldsymbol{\delta}) \right\}$, $\overline{X}^p(\boldsymbol{\delta}) = X_p \cap \overline{X}(\boldsymbol{\delta})$ and $\overline{X}^q(\boldsymbol{\delta}) = X_q \cap \overline{X}(\boldsymbol{\delta})$. See Figure 1b for a visualization of $t(\boldsymbol{\delta})$ and $\overline{X}(\boldsymbol{\delta})$ (the dark shaded region).

**Setting:** Suppose dataset $X_p \overset{\text{i.i.d.}}{\sim} P$ and $X_q \overset{\text{i.i.d.}}{\sim} Q$. Truncated densities $\overline{p}_{\boldsymbol{\delta}}$ and $\overline{q}_{\boldsymbol{\delta}}$ are the unbounded densities $p$ and $q$ restricted only on the truncated domain $\overline{X}(\boldsymbol{\delta})$. Note that the truncated densities are dependent on the parameter $\boldsymbol{\delta}$ and $\nu$. We show that under some assumptions, the parameter $\hat{\boldsymbol{\delta}}$ obtained from (9) using a fixed hyperparameter $\nu$ will converge to the $\boldsymbol{\delta}^*$ such that $\overline{q}_{\boldsymbol{\delta}^*}(\boldsymbol{x})r(\boldsymbol{x}; \boldsymbol{\delta}^*) = \overline{p}_{\boldsymbol{\delta}^*}(\boldsymbol{x})$. We also define the "optimal" weight assignment $w_i^* = \frac{1}{n_p}, \forall i, \boldsymbol{x}_p^{(i)} \in \overline{X}(\boldsymbol{\delta}^*)$ and 0 otherwise. Interestingly, the constraint in (9), $\langle \boldsymbol{w}^*, \boldsymbol{1} \rangle = \nu$ may *not* hold, but our analysis in this section suggests we can always find a pair $(\hat{\boldsymbol{\delta}}, \hat{\boldsymbol{w}})$ in the feasible region so that $\|\hat{\boldsymbol{\delta}} - \boldsymbol{\delta}^*\|$ converges to 0 under mild conditions.

We first assume the log density ratio model and its CDF is Lipschitz continuous.

**Assumption 4.**
$$\forall \boldsymbol{u} \in \mathrm{Ball}(\rho), \sup_{\boldsymbol{x}} |\hat{z}_{\boldsymbol{\delta}^* + \boldsymbol{u}}(\boldsymbol{x}) - \hat{z}_{\boldsymbol{\delta}^*}(\boldsymbol{x})| \leq C_{\mathrm{lip}}\|\boldsymbol{u}\|. \tag{12}$$

*Define* $T(\boldsymbol{u}, \epsilon) := \{\boldsymbol{x} \in \mathbb{R}^d \mid |z_{\boldsymbol{\delta}^*}(\boldsymbol{x}) - t(\boldsymbol{\delta}^*)| \le 2C_{\text{lip}}\|\boldsymbol{u}\| + \epsilon\}$ *where* $0 < \epsilon \le 1$. *We assume* $\forall \boldsymbol{u} \in \text{Ball}(\rho), 0 < \epsilon \le 1$

$$P\left[\boldsymbol{x}_p \in T(\boldsymbol{u}, \epsilon)\right] \le C_{\text{CDF}} \cdot \|\boldsymbol{u}\| + \epsilon.$$

In this assumption, we define a "zone" $T(\boldsymbol{u}, \epsilon)$ near the $\nu$-th quantile $t(\boldsymbol{\delta}^*)$ and assume the CDF of our ratio model is upper-bounded over this region. Different from Assumption 3, the RHS of (12) is with respect to $\ell_2$ norm of $\boldsymbol{u}$. In the following assumption, we assume regularity on $P$ and $Q$.

**Assumption 5.** $\forall \boldsymbol{x}_q \in \mathbb{R}^d, \|\boldsymbol{f}(\boldsymbol{x}_q)\|_\infty \le C_q$ *and* $\forall \boldsymbol{u} \in \text{Ball}(\rho), \forall \boldsymbol{x}_p \in T(\boldsymbol{u}, 1), \|\boldsymbol{f}(\boldsymbol{x}_p)\|_\infty \le C_p$.

**Theorem 3.** *In addition Assumption 1 and 2 and other mild assumptions (see Section 8.6 in the supplementary material), Assumption 4 and 5 hold. If* $1 \ge \nu \ge \frac{8C_{\text{CDF}}\sqrt{k}C_pC_r^2}{\kappa_1'}, n_q = \Omega(|\overline{X}^p(\boldsymbol{\delta}^*)|^2)$,

$$\lambda_n \ge 2 \max\left[\sqrt{\frac{K_1' \log d}{|\overline{X}^p(\boldsymbol{\delta}^*)|}} + \frac{2C_r^2 C_q |X_q \backslash \overline{X}^q(\boldsymbol{\delta}^*)|}{n_q}, \frac{2L \cdot C_p}{\sqrt{n_p}}, \frac{\rho\nu c}{2C_r^2\sqrt{n_q}}\right],$$

*where* $K_1' > 0, c > 0$ *are constants, we are guaranteed that* $\|\hat{\boldsymbol{\delta}} - \boldsymbol{\delta}^*\| \le \frac{4C_r^2}{\nu\kappa_1'} \cdot 3\sqrt{k}\lambda_n$ *with high probability.*

It can be seen that $\|\hat{\boldsymbol{\delta}} - \boldsymbol{\delta}^*\| = O\left(\sqrt{\log d / \min(|\overline{X}^p(\boldsymbol{\delta}^*)|, n_q)}\right)$ if $d$ is reasonably large and $|X_q \backslash \overline{X}^q(\boldsymbol{\delta}^*)|/n_q$ decays fast.

# 6 Experiments

## 6.1 Detecting Sparse Structural Changes between Two Markov Networks (MNs) [14]

In the first experiment[3], we learn changes between two Gaussian MNs under the outlier setting. The ratio between two Gaussian MNs can be parametrized as $p(\boldsymbol{x})/q(\boldsymbol{x}) \propto \exp(-\sum_{i,j \le d} \Delta_{i,j} x_i x_j)$, where $\Delta_{i,j} := \Theta_{i,j}^p - \Theta_{i,j}^q$ is the difference between precision matrices. We generate 500 samples as $X_p$ and $X_q$ using two randomly structured Gaussian MNs. One point $[10, \ldots, 10]$ is added as an outlier to $X_p$. To induce sparsity, we set $R(\boldsymbol{\Delta}) = \sum_{i,j=1,i \le j}^d |\Delta_{i,j}|$ and fix $\lambda = 0.0938$. Then run DRE and TRimmed-DRE to learn the sparse *differential* precision matrix $\boldsymbol{\Delta}$ and results are plotted on Figure 2a and 2b[4] where the ground truth (the position $i, j, \Delta_{i,j}^* \ne 0$) is marked by red boxes. It can be seen that the outlier completely misleads DRE while TR-DRE performs reasonably well. We also run experiments with two different settings ($d = 25, d = 36$) and plot True Negative Rate (TNR) - True Positive Rate (TPR) curves. We fix $\nu$ in TR-DRE to 90% and compare the performance of DRE and TR-DRE using DRE without any outliers as gold standard (see Figure 2c). It can be seen that the added outlier makes the DRE fail completely while TR-DRE can almost reach the gold standard. It also shows the price we pay: TR-DRE does lose some power for discarding samples. However, the loss of performance is still acceptable.

## 6.2 Relative Novelty Detection from Images

In the second experiment, we collect four images (see Figure 3a) containing three objects with a textured background: a pencil, an earphone and an earphone case. We create data points from these four images using sliding windows of $48 \times 48$ pixels (the green box on the lower right picture on Figure 3a). We extract 899 features using MATLAB HOG method on each window and construct an 899-dimensional sample. Although our theorems in Section 5 are proved for linear models, here $\boldsymbol{f}(\boldsymbol{x})$ is an RBF kernel using all samples in $X_p$ as kernel basis. We pick the top left image as $X_p$ and using all three other images as $X_q$, then run TR-DRE, THresholded-DRE [26], and one-SVM.

In this task, we select high density ratio super pixels on image $X_p$. It can be expected that the super pixels containing the pencil will exhibit high density ratio values as they did not appear in the reference dataset $X_q$ while super pixels containing the earphone case, the earphones and the background, repeats similar patches in $X_q$ will have lower density ratio values. This is different from

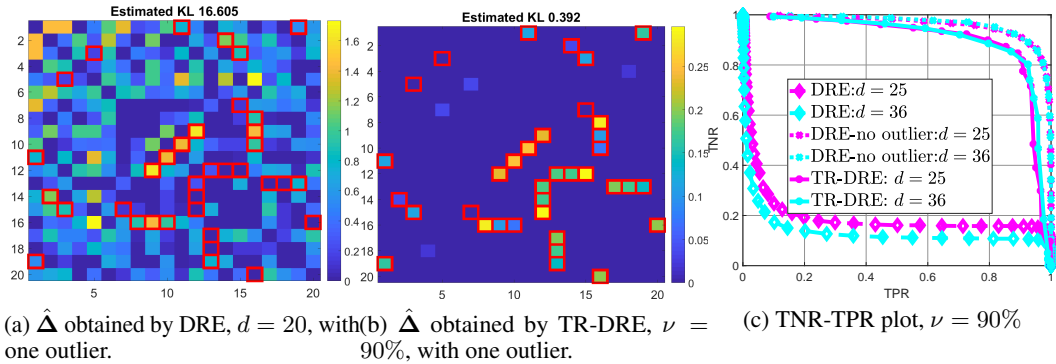

(a) $\hat{\mathbf{\Delta}}$ obtained by DRE, $d = 20$, with one outlier.    (b) $\hat{\mathbf{\Delta}}$ obtained by TR-DRE, $\nu = 90\%$, with one outlier.    (c) TNR-TPR plot, $\nu = 90\%$

Figure 2: Using DRE to learn changes between two MNs. We set $R(\cdot) = \|\cdot\|_1$ and $f(x_i, x_j) = x_i x_j$.

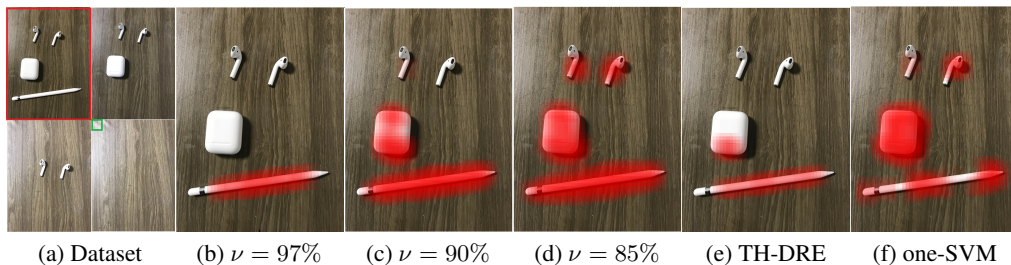

(a) Dataset    (b) $\nu = 97\%$    (c) $\nu = 90\%$    (d) $\nu = 85\%$    (e) TH-DRE    (f) one-SVM

Figure 3: Relative object detection using super pixels. We set $R(\cdot) = \|\cdot\|^2$, $\boldsymbol{f}(\boldsymbol{x})$ is an RBF kernel.

a conventional novelty detection, as a density ratio function help us capture only the relative novelty. For TR-DRE, we use the trimming threshold $\hat{t}$ as the threshold for selecting high density ratio points.

It can be seen on Figure 3b, 3c and 3d, as we tune $\nu$ to allow more and more high density ratio windows to be selected, more relative novelties are detected: First the pen, then the case, and finally the earphones, as the lack of appearance in the reference dataset $X_q$ elevates the density ratio value by different degrees. In comparison, we run TH-DRE with top 3% highest density ratio values thresholded, which corresponds to $\nu = 97\%$ in our method. The pattern of the thresholded windows (shaded in red) in Figure 3e is similar to Figure 3b though some parts of the case are mistakenly shaded. Finally, one-SVM with 3% support vectors (see Figure 3f) does not utilize the knowledge of a reference dataset $X_q$ and labels all salient objects in $X_p$ as they corresponds to the "outliers" in $X_p$.

## 7 Conclusion

We presents a robust density ratio estimator based on the idea of trimmed MLE. It has a convex formulation and the optimization can be easily conducted using a subgradient ascent method. We also investigate its theoretical property through an equivalent weighted M-estimator whose $\ell_2$ estimation error bound was provable under two high-dimensional, robust settings. Experiments confirm the effectiveness and robustness of the our trimmed estimator.

## Acknowledgments

We thank three anonymous reviewers for their detailed and helpful comments. Akiko Takeda thanks Grant-in-Aid for Scientific Research (C), 15K00031. Taiji Suzuki was partially supported by MEXT KAKENHI (25730013, 25120012, 26280009 and 15H05707), JST-PRESTO and JST-CREST. Song Liu and Kenji Fukumizu have been supported in part by MEXT Grant-in-Aid for Scientific Research on Innovative Areas (25120012).

## Footnotes

[2] https://www.tensorflow.org/

[3]Code can be found at http://allmodelsarewrong.org/software.html

[4]Figures are best viewed in color.

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
