[Supplementary Material · supp.pdf]

# 8 Appendix

To lighten the notation system, we drop the feature transform $\boldsymbol{f}$ from our equations. The analysis procedure does not change with or without $\boldsymbol{f}$.

## 8.1 Relationship between Trimmed DRE and Binary SVM [23, 4]

Consider a "symmetrized" extension to the criterion (6):

$$\min_{\boldsymbol{\delta}} \mathrm{KL}\left[p|q \cdot r_{\boldsymbol{\delta}}\right] + \mathrm{KL}\left[q|p \cdot 1/r_{\boldsymbol{\delta}}\right]$$

$$\approx c - \max_{\boldsymbol{\delta}} \frac{1}{n_p} \sum_{i=1}^{n_p} \log \hat{r}(\boldsymbol{x}_p^{(i)}; \boldsymbol{\delta}) + \frac{1}{n_q} \sum_{i=1}^{n_q} \log \hat{r}_2(\boldsymbol{x}_q^{(i)}; \boldsymbol{\delta}) \tag{13}$$

that jointly minimizes the KL divergence from $P$ to $Q$ and from $Q$ to $P$. Similar to (5), we use $\hat{r}_2$ to model the ratio $q/p$:

$$\hat{r}_2(\boldsymbol{x}; \boldsymbol{\delta}) = \frac{\exp\langle -\boldsymbol{\delta}, \boldsymbol{x}\rangle}{\hat{N}_2(\boldsymbol{\delta})}, \hat{N}_2(\boldsymbol{\delta}) := \frac{1}{n_p} \sum_{j=1}^{n_p} \exp\langle -\boldsymbol{\delta}, \boldsymbol{x}_p^{(j)}\rangle.$$

The minus in front of the $\boldsymbol{\delta}$ is due to the inversion of the ratio. We can trim the objective function (13) and add a regularization term $\lambda R(\boldsymbol{\delta})$ as we did for the asymmetric one:

$$\max_{\boldsymbol{\delta}} \frac{1}{n_p} \sum_{i=1}^{n_p} [\log \hat{r}(\boldsymbol{x}_p^{(i)}; \boldsymbol{\delta}) - t_0]_- + \frac{1}{n_q} \sum_{i=1}^{n_q} [\log \hat{r}_2(\boldsymbol{x}_q^{(i)}; \boldsymbol{\delta}) - t_0]_- - \lambda R(\boldsymbol{\delta}) \tag{14}$$

**Proposition 2.** *If $n_p = n_q, t_0 = 1, R(\cdot) = \|\cdot\|_2^2$, the maximizer $\hat{\boldsymbol{\delta}}$ of (14) is the same as the primal solution of a modified SVM using $X_p$ and $X_q$ as positive and negative class respectively.*

It suggests SVM learns an *unnormalized* and *trimmed* density ratio function as the decision function.

*Proof.* By introducing the slack variables as we did in (7). (14) can be rewritten as:

$$\min_{\boldsymbol{\delta}, \boldsymbol{\epsilon} \geq \mathbf{0}} \frac{1}{n_p}\langle \epsilon_p, \mathbf{1}\rangle + \frac{1}{n_q}\langle \epsilon_q, \mathbf{1}\rangle + \lambda R(\boldsymbol{\delta})$$

$$\text{s.t. } \forall \boldsymbol{x}_p^{(i)} \in X_p, \forall \boldsymbol{x}_q^{(i)} \in X_q,$$

$$\log \hat{r}(\boldsymbol{x}_p^{(i)}; \boldsymbol{\delta}) \geq t_0 - \epsilon_{p,i},$$

$$\log \hat{r}_2(\boldsymbol{x}_q^{(i)}; \boldsymbol{\delta}) \geq t_0 - \epsilon_{q,i}, \tag{15}$$

After substituting $\hat{r}$ and $\hat{r}_2$, (15) can be re-written as

$$\min_{\boldsymbol{\delta}, \boldsymbol{\epsilon} \geq \mathbf{0}} \frac{1}{n_p}\langle \epsilon_p, \mathbf{1}\rangle + \frac{1}{n_q}\langle \epsilon_q, \mathbf{1}\rangle + \lambda R(\boldsymbol{\delta})$$

$$\text{s.t. } \forall \boldsymbol{x}_p^{(i)} \in X_p, \forall \boldsymbol{x}_q^{(i)} \in X_q,$$

$$\langle \boldsymbol{\delta}, \boldsymbol{x}_p^{(i)}\rangle - \log \hat{N}(\boldsymbol{\delta}) \geq t_0 - \epsilon_{p,i},$$

$$\langle -\boldsymbol{\delta}, \boldsymbol{x}_q^{(i)}\rangle - \log \hat{N}_2(\boldsymbol{\delta}) \geq t_0 - \epsilon_{q,i}. \tag{16}$$

Let $n_p = n_q, t_0 = 1, R(\boldsymbol{\delta}) = \|\boldsymbol{\delta}\|^2$, (16) is an SVM (without a bias term) using $X_p$ and $X_q$ as positive and negative samples respectively, except the presences of log normalization terms $\log \hat{N}(\boldsymbol{\delta})$ and $\log \hat{N}_2(\boldsymbol{\delta})$. □

## 8.2 Proof of Proposition 1

*Proof.* To prove the statement, we construct the dual of (8) which has the exactly same form as (9). Denote $\boldsymbol{X}_p = \left[\boldsymbol{x}_p^{(1)}, \ldots, \boldsymbol{x}_p^{(n_p)}\right] \in \mathbb{R}^{d \times n_p}$ and $\boldsymbol{X}_q = \left[\boldsymbol{x}_q^{(1)}, \ldots, \boldsymbol{x}_q^{(n_q)}\right] \in \mathbb{R}^{d \times n_q}$.

The Lagrangian of (8) can be written as

$$l(\boldsymbol{\alpha}, \boldsymbol{\alpha}', \alpha'', \boldsymbol{\delta}, t, \boldsymbol{\epsilon}) = -\langle \boldsymbol{\alpha}, \boldsymbol{\delta}^\top \boldsymbol{X}_p - \log \widehat{N}(\boldsymbol{\delta}) \cdot \mathbf{1} - t \cdot \mathbf{1} + \boldsymbol{\epsilon} \rangle$$

$$-\langle \boldsymbol{\alpha}', \boldsymbol{\epsilon} \rangle - \alpha'' \cdot t + \frac{1}{n_p} \langle \boldsymbol{\epsilon}, \mathbf{1} \rangle - \nu \cdot t + \lambda R(\boldsymbol{\delta}) \tag{17}$$

where $\boldsymbol{\alpha} \in \mathbb{R}_+^{n_p}, \boldsymbol{\alpha}' \in \mathbb{R}_+, \alpha'' \in \mathbb{R}_+$. Now we analyze the KKT condition of the above Lagrangian.

Suppose the optimal $\hat{t} > 0$[5], then $\alpha'' = 0$ by the slackness condition that $t'\alpha'' = 0$. The optimality condition of $t$ in (17) yields:

$$\nabla_t l(\boldsymbol{\alpha}, \boldsymbol{\alpha}', \alpha'', \boldsymbol{\delta}, t, \boldsymbol{\epsilon}) = \langle \boldsymbol{\alpha}, \mathbf{1} \rangle - \nu = 0 \rightarrow \sum_{i=1}^{n_p} \alpha_i = \nu, \tag{18}$$

and the optimality condition of $\boldsymbol{\epsilon}$ yields

$$\nabla_{\boldsymbol{\epsilon}} l(\boldsymbol{\alpha}, \boldsymbol{\alpha}', \alpha'', \boldsymbol{\delta}, t, \boldsymbol{\epsilon}) = \mathbf{0} \rightarrow -\boldsymbol{\alpha} - \boldsymbol{\alpha}' + \frac{1}{n_p} \cdot \mathbf{1} = \mathbf{0} \tag{19}$$

From (18) and (19), and the slackness condition of optimization (7), we can see $\boldsymbol{x}_p^{(i)} \in \boldsymbol{X}_p$, if $\log \hat{r}(\boldsymbol{x}_p^{(i)}; \boldsymbol{\delta}) < t$, then $\epsilon_i > 0$ which leads to $\alpha_i' = 0$ (the constraint of $\epsilon_i \geq 0$ is ineffective) and thus $\alpha_i = \frac{1}{n_p}$.

In contrast, if $\log \hat{r}(\boldsymbol{x}_p^{(i)}; \boldsymbol{\delta}) > t$, then we have $\alpha_i = 0, \epsilon_i = 0$ (the constraint of $\epsilon_i \geq 0$ is *effective*). If $\log \hat{r}(\boldsymbol{x}_p^{(i)}; \boldsymbol{\delta})$ falls right on the boundary $t$, i.e., $\log r(\boldsymbol{x}_p^{(i)}; \boldsymbol{\delta}) = t$, $\alpha_i \in [0, \frac{1}{n_p}]$, since the KKT condition $\epsilon_i \alpha_i' = 0$ indicating $\alpha_i'$ can take non-negative values as long as $\frac{1}{n_p} \cdot \mathbf{1} = \boldsymbol{\alpha} + \boldsymbol{\alpha}'$. We summarize:

$$\begin{cases} \alpha_i = \frac{1}{n_p} & \log \hat{r}(\boldsymbol{x}_p^{(i)}; \boldsymbol{\delta}) < t \\ 0 \leq \alpha_i \leq \frac{1}{n_p} & \log \hat{r}(\boldsymbol{x}_p^{(i)}; \boldsymbol{\delta}) = t \\ \alpha_i = 0 & \log \hat{r}(\boldsymbol{x}_p^{(i)}; \boldsymbol{\delta}) > t. \end{cases} \tag{20}$$

It can be observed that for (8), $(\boldsymbol{\delta} = \mathbf{0}, \boldsymbol{\epsilon} = 0.2 \cdot \mathbf{1}, t = 0.1)$ is a feasible interior point, and it makes all inequality constraints strict, so the *Slater's condition* holds for our original primal problem which is also convex. Therefore, the lagrangian dual of the original problem (8) is

$$\min_{\boldsymbol{\delta}} \max_{\boldsymbol{\alpha} \geq 0, \boldsymbol{\alpha}' \geq 0, \alpha'' \geq 0} \min_{\boldsymbol{\epsilon}, t} l(\boldsymbol{\alpha}, \boldsymbol{\alpha}', \alpha'', \boldsymbol{\delta}, t, \boldsymbol{\epsilon})$$

$$= \min_{\boldsymbol{\delta}} \max_{\boldsymbol{\alpha}} -\langle \boldsymbol{\alpha}, \boldsymbol{\delta}^\top \boldsymbol{X}_p - \log \widehat{N}(\boldsymbol{\delta}) \rangle + \lambda R(\boldsymbol{\delta}) \tag{21}$$

$$s.t. \boldsymbol{\alpha} \in \left[0, \frac{1}{n_p}\right]^{n_p}, \langle \mathbf{1}, \boldsymbol{\alpha} \rangle = \nu. \tag{22}$$

which is the same as (9) and any points satisfy the KKT condition are both dual (22) and primal (8) optimal. □

## 8.3 Lemma 1

**Lemma 1.** *If Assumptions 1 and 2 hold, then*

$$-\boldsymbol{u}^\top \nabla_{\boldsymbol{\delta}}^2 \mathcal{L}(\boldsymbol{\delta}^* + \boldsymbol{u}, \boldsymbol{w}^*) \boldsymbol{u} \geq \frac{\nu \kappa_1'}{2 C_r^2} \|\boldsymbol{u}\|^2 - \frac{\nu c}{2 C_r^2} \cdot \frac{\|\boldsymbol{u}\|_1^2}{\sqrt{n_q}}, \tag{23}$$

where $c$ is the constant determined by Assumption 2.

*Proof.* First, we write down $-\nabla_{\boldsymbol{\delta}}^2 \mathcal{L}(\boldsymbol{\delta}^* + \boldsymbol{u}, \boldsymbol{w}^*)$:

$$-\nabla_{\boldsymbol{\delta}}^2 \mathcal{L}(\boldsymbol{\delta}^* + \boldsymbol{u}, \boldsymbol{w}^*) = -\nabla^2 \sum_{i=1}^{n_p} w_i^* \cdot \log \hat{r}(\boldsymbol{x}_p^{(i)}; \boldsymbol{\delta}^* + \boldsymbol{u})$$

$$= -\sum_{i=1}^{n_p} w_i^* \cdot \nabla^2 \log \widehat{N}(\boldsymbol{\delta}^* + \boldsymbol{u})$$

$$= -\nu \cdot \nabla^2 \log \widehat{N}(\boldsymbol{\delta}^* + \boldsymbol{u}),$$

$$= \nu \cdot \sum_{i=1}^{n_q} \frac{e^{(i)}}{s} \cdot \boldsymbol{x}_q^{(i)} \left(\boldsymbol{x}_q^{(i)}\right)^\top - \nu \cdot \left\{\sum_{i=1}^{n_q} \frac{e^{(i)}}{s} \cdot \left(\boldsymbol{x}_q^{(i)}\right)\right\} \left\{\sum_{i=1}^{n_q} \frac{e^{(i)}}{s} \cdot \left(\boldsymbol{x}_q^{(i)}\right)\right\}^\top$$

where $e^{(j)} := \exp\left[\langle \boldsymbol{\delta}^* + \boldsymbol{u}, \boldsymbol{x}^{(j)}\rangle\right], s := \sum_{j=1}^{n_q} e^{(j)}$.

$$\nu \boldsymbol{u}^\top \left\{\sum_{i=1}^{n_q} \frac{e^{(i)}}{s} \cdot \boldsymbol{x}^{(i)} \left(\boldsymbol{x}^{(i)}\right)^\top - \left\{\sum_{i=1}^{n_q} \frac{e^{(i)}}{s} \cdot \left(\boldsymbol{x}^{(i)}\right)\right\} \left\{\sum_{i=1}^{n_q} \frac{e^{(i)}}{s} \cdot \left(\boldsymbol{x}^{(i)}\right)\right\}^\top\right\} \boldsymbol{u}$$

$$= \frac{\nu}{2} \boldsymbol{u}^\top \left\{\sum_{i=1}^{n_q} \sum_{j \neq i} \frac{e^{(i)} e^{(j)}}{s^2} \left(\boldsymbol{x}^{(i)} - \boldsymbol{x}^{(j)}\right)\left(\boldsymbol{x}^{(i)} - \boldsymbol{x}^{(j)}\right)^\top\right\} \boldsymbol{u}$$

Due to Assumption 1, $\frac{e^{(i)}}{s} \geq \frac{1}{C_r n_q}$. Let $\xi_{i,j} = \left(\boldsymbol{x}^{(i)} - \boldsymbol{x}^{(j)}\right)\left(\boldsymbol{x}^{(i)} - \boldsymbol{x}^{(j)}\right)^\top$, then we have the following inequalities

$$\frac{\nu}{2} \boldsymbol{u}^\top \left\{\sum_{i=1}^{n_q} \sum_{j \neq i} \frac{e^{(i)} e^{(j)}}{s^2} \xi_{i,j}\right\} \boldsymbol{u} \geq \frac{\nu}{2C_r^2} \boldsymbol{u}^\top \left\{\frac{1}{n_q^2} \sum_{i=1}^{n_q} \sum_{j \neq i} \xi_{i,j}\right\} \boldsymbol{u} = \frac{\nu}{2C_r^2} \boldsymbol{u}^\top \text{cov}(\boldsymbol{X}_q) \boldsymbol{u}$$

We then invoke Assumption 2 to obtain $\frac{\nu}{2C_r^2} \boldsymbol{u}^\top \text{cov}(\boldsymbol{X}_q) \boldsymbol{u} \geq \frac{\nu \kappa_1'}{2C_r^2} \|\boldsymbol{u}\|^2 - \frac{\nu c}{2C_r^2 \sqrt{n_q}} \|\boldsymbol{u}\|_1^2$. □

## 8.4 Proof of Theorem 1

*Proof.* First, we define the $S$ and $S^c$ are the set of indices of non-zero and zero elements of $\boldsymbol{\delta}^*$. The cardinlity of $S$ is $k$.

Define $\hat{\boldsymbol{u}} := \hat{\boldsymbol{\delta}} - \boldsymbol{\delta}^*$. From the Lemma 1 we can see that,

$$\langle \nabla_{\boldsymbol{\delta}} \mathcal{L}(\hat{\boldsymbol{\delta}}, \boldsymbol{w}^*) - \nabla_{\boldsymbol{\delta}} \mathcal{L}(\boldsymbol{\delta}^*, \boldsymbol{w}^*), \hat{\boldsymbol{u}}\rangle \geq \kappa_1 \|\hat{\boldsymbol{u}}\|^2 - \tau_1(n, d)\|\hat{\boldsymbol{u}}\|_1^2,$$

where we set $\kappa_1 := \frac{\nu \kappa_1'}{2C_r^2}, \tau_1(n, d) := \frac{\nu c}{2C_r^2 \sqrt{n_q}}$. Using Holder's inequality,

$$\langle \nabla_{\boldsymbol{\delta}} \mathcal{L}(\hat{\boldsymbol{\delta}}, \boldsymbol{w}^*), \hat{\boldsymbol{u}}\rangle + \|\nabla_{\boldsymbol{\delta}} \mathcal{L}(\boldsymbol{\delta}^*, \boldsymbol{w}^*)\|_\infty \|\hat{\boldsymbol{u}}\|_1 + \tau_1(n, d)\rho\|\hat{\boldsymbol{u}}\|_1 \geq \kappa_1 \|\hat{\boldsymbol{u}}\|^2.$$

The introduction of $\rho$ is due to the bounded optimization region. Due to (11), we can convert the above inequality into

$$\langle \nabla_{\boldsymbol{\delta}} \mathcal{L}(\hat{\boldsymbol{\delta}}, \hat{\boldsymbol{w}}), \hat{\boldsymbol{u}}\rangle + \kappa_2 \|\hat{\boldsymbol{u}}\|^2 + \tau_2(n, d)\|\hat{\boldsymbol{u}}\|_1 + \|\nabla_{\boldsymbol{\delta}} \mathcal{L}(\boldsymbol{\delta}^*, \boldsymbol{w}^*)\|_\infty \|\hat{\boldsymbol{u}}\|_1 + \rho \tau_1(n, d)\|\hat{\boldsymbol{u}}\|_1 \geq \kappa_1 \|\hat{\boldsymbol{u}}\|^2,$$

and because of the setting of $\lambda_n$,

$$\langle \nabla_{\boldsymbol{\delta}} \mathcal{L}(\hat{\boldsymbol{\delta}}, \hat{\boldsymbol{w}}), \hat{\boldsymbol{u}}\rangle + \frac{\lambda_n}{2} \|\hat{\boldsymbol{u}}\|_1 \geq (\kappa_1 - \kappa_2)\|\hat{\boldsymbol{u}}\|^2, \tag{24}$$

Note that in the first term, $\hat{\boldsymbol{\delta}}$ is obtained at the stationary condition, which implies that there is a subgradient, denoted by $\nabla \|\hat{\boldsymbol{\delta}}\|_1$, such that

$$\nabla_{\boldsymbol{\delta}} \mathcal{L}(\hat{\boldsymbol{\delta}}, \hat{\boldsymbol{w}}) = -\lambda_n \nabla_{\boldsymbol{\delta}} \|\hat{\boldsymbol{\delta}}\|_1 = -\lambda_n \nabla_{\boldsymbol{\delta}} \|\hat{\boldsymbol{u}} + \boldsymbol{\delta}^*\|_1,$$

(the second $\nabla$ is the subgradient notation) thus we can obtain the upper-bound of $\langle \nabla_{\boldsymbol{\delta}}\mathcal{L}(\hat{\boldsymbol{\delta}}, \hat{\boldsymbol{w}}), \hat{\boldsymbol{u}} \rangle$ using the following standard procedure:

$$
\begin{aligned}
\langle \nabla_{\boldsymbol{\delta}}\mathcal{L}(\hat{\boldsymbol{\delta}}, \hat{\boldsymbol{w}}), \hat{\boldsymbol{u}} \rangle &= -\lambda_n \langle \nabla_{\boldsymbol{\delta}} \|\hat{\boldsymbol{u}} + \boldsymbol{\delta}^*\|_1, \hat{\boldsymbol{u}} \rangle \\
&\leq -\lambda_n(\|\hat{\boldsymbol{\delta}}\|_1 - \|\boldsymbol{\delta}^*\|_1) \text{ due to convexity of } \|\boldsymbol{\delta}\|_1 \text{ and the definition of subgradient.} \\
&= \lambda_n(\|\boldsymbol{\delta}^*\|_1 + \|\hat{\boldsymbol{u}}_{S^c}\|_1 - \|\hat{\boldsymbol{u}}_{S^c}\|_1 - \|\hat{\boldsymbol{\delta}}\|_1) \\
&= \lambda_n(\|\boldsymbol{\delta}^* + \hat{\boldsymbol{u}}_{S^c}\|_1 - \|\hat{\boldsymbol{u}}_{S^c}\|_1 - \|\hat{\boldsymbol{\delta}}\|_1) \\
&= \lambda_n(\|\boldsymbol{\delta}^* + \hat{\boldsymbol{u}}_{S^c}\|_1 + \|\hat{\boldsymbol{u}}_S\|_1 - \|\hat{\boldsymbol{u}}_S\|_1 - \|\hat{\boldsymbol{u}}_{S^c}\|_1 - \|\hat{\boldsymbol{\delta}}\|_1) \\
&\leq \lambda_n(\|\boldsymbol{\delta}^* + \hat{\boldsymbol{u}}_S + \hat{\boldsymbol{u}}_{S^c}\|_1 + \|\hat{\boldsymbol{u}}_S\|_1 - \|\hat{\boldsymbol{u}}_{S^c}\|_1 - \|\hat{\boldsymbol{\delta}}\|_1) \\
&\leq \lambda_n(\|\hat{\boldsymbol{u}}_S\|_1 - \|\hat{\boldsymbol{u}}_{S^c}\|_1) 
\end{aligned}
\tag{25}
$$

Combining (24) and (25) we have

$$
\lambda_n(\|\hat{\boldsymbol{u}}_S\|_1 - \|\hat{\boldsymbol{u}}_{S^c}\|_1) + \frac{\lambda_n}{2}\|\hat{\boldsymbol{u}}\|_1 \geq (\kappa_1 - \kappa_2)\|\hat{\boldsymbol{u}}\|^2
$$

$$
\frac{3\lambda_n}{2}\|\hat{\boldsymbol{u}}_S\|_1 - \frac{\lambda_n}{2}\|\hat{\boldsymbol{u}}_{S^c}\|_1 \geq (\kappa_1 - \kappa_2)\|\hat{\boldsymbol{u}}\|^2
\tag{26}
$$

$$
\frac{3\lambda_n\sqrt{k}}{2}\|\hat{\boldsymbol{u}}\|_2 \geq (\kappa_1 - \kappa_2)\|\hat{\boldsymbol{u}}\|^2
$$

$$
\frac{1}{(\kappa_1 - \kappa_2)} \cdot \frac{3\sqrt{k}\lambda_n}{2} \geq \|\hat{\boldsymbol{u}}\|.
$$

Substituting $\kappa_1$ and $\tau_1(n, d)$ according to Lemma 1, we have the conclusion in Theorem 1. $\qquad \square$

## 8.5 Proof of Theorem 2

Now let's specify $\kappa_2$ and $\tau_2$ in Theorem 1 under the outlier setting and derive the consistency.

Let's consider (11). It is easy to see that

$$
\nabla_{\boldsymbol{\delta}}\mathcal{L}(\hat{\boldsymbol{\delta}}, \hat{\boldsymbol{w}}) - \nabla_{\boldsymbol{\delta}}\mathcal{L}(\hat{\boldsymbol{\delta}}, \boldsymbol{w}^*) = \sum_{i\in\hat{G}} w_i \boldsymbol{f}(\boldsymbol{x}_p^{(i)}) - \frac{1}{n_p}\sum_{i\in G} \boldsymbol{f}(\boldsymbol{x}_p^{(i)}), \text{ where } \hat{G} := \{\boldsymbol{x}_p^{(i)}|\hat{w}_i \neq 0\}.
$$

It is obvious that if $\hat{G} \equiv G$ and $\forall i \in \hat{G}, \hat{w}_i = \frac{1}{n_p}$, and $\forall i \in B, \hat{w}_i = 0$, $\nabla_{\boldsymbol{\delta}}\mathcal{L}(\hat{\boldsymbol{\delta}}, \hat{\boldsymbol{w}}) - \nabla_{\boldsymbol{\delta}}\mathcal{L}(\hat{\boldsymbol{\delta}}, \boldsymbol{w}^*) = 0$.

**Lemma 2.** *If there exists a "clearance" between the good samples and the bad samples, such that* $\min_{j\in B} z_{\boldsymbol{\delta}^*}(\boldsymbol{x}_p^{(j)}) - \max_{i\in G} z_{\boldsymbol{\delta}^*}(\boldsymbol{x}_p^{(i)}) \geq 3C_{\text{lip}}\rho$, *then* $\nabla_{\boldsymbol{\delta}}\mathcal{L}(\hat{\boldsymbol{\delta}}, \hat{\boldsymbol{w}}) - \nabla_{\boldsymbol{\delta}}\mathcal{L}(\hat{\boldsymbol{\delta}}, \boldsymbol{w}^*) = 0$.

*Proof.*

$$
\min_{j\in B} z_{\boldsymbol{\delta}^*}(\boldsymbol{x}_p^{(j)}) - \max_{i\in G} z_{\boldsymbol{\delta}^*}(\boldsymbol{x}_p^{(i)}) = \min_{j\in B} \hat{z}_{\boldsymbol{\delta}^*}(\boldsymbol{x}^{(j)}) - \max_{i\in G} \hat{z}_{\boldsymbol{\delta}^*}(\boldsymbol{x}^{(i)}) \geq 3C_{\text{lip}}\rho
\tag{27}
$$

Due to Assumption 3 and (27),

$$
\forall i \in G, j \in B, \text{and } \boldsymbol{u} \in \text{Ball}(\rho), \ \hat{z}_{\boldsymbol{\delta}^*+\boldsymbol{u}}(\boldsymbol{x}^{(j)}) > \hat{z}_{\boldsymbol{\delta}^*+\boldsymbol{u}}(\boldsymbol{x}^{(i)}).
\tag{28}
$$

According to the optimality condition of (9), we should simply assign non-zero weights $w_i$ to the $\nu n_p$ samples corresponding to the smallest $\hat{z}_{\boldsymbol{\delta}^*+\boldsymbol{u}}$ values. Therefore, from (28) we can see that $\hat{G} = G$. Moreover, since the inequality of (28) holds strictly and $\nu = \frac{|G|}{n_p} = \frac{|\hat{G}|}{n_p}$, all weights must be set to $\frac{1}{n_p}$ in order to minimize the inner problem of (9), i.e., $\forall i \in G, \hat{w}_i = \frac{1}{n_p}$ and $\forall i \in B, \hat{w}_i = 0$. $\qquad \square$

Now we can set $\kappa_2 = 0, \tau_2(n, d) = 0$ to make (11) hold.

As explained in Section (5.1), we need to confirm $\|\nabla_{\boldsymbol{\delta}}\mathcal{L}(\boldsymbol{\delta}^*, \boldsymbol{w}^*)\|_\infty$ converges to 0 as the sample size goes to inifinity where $\nabla_{\boldsymbol{\delta}}\mathcal{L}(\boldsymbol{\delta}^*, \boldsymbol{w}^*) = \frac{1}{n_p}\sum_{i\in G} \nabla_{\boldsymbol{\delta}}\hat{z}_{\boldsymbol{\delta}^*}(\boldsymbol{x}_p^{(i)})$. Since

$$
\|\frac{1}{n_p}\sum_{i\in G} \nabla_{\boldsymbol{\delta}}\hat{z}_{\boldsymbol{\delta}^*}(\boldsymbol{x}_p^{(i)})\|_\infty \leq \frac{1}{\nu}\cdot\|\frac{1}{n_p}\sum_{i\in G} \nabla_{\boldsymbol{\delta}}\hat{z}_{\boldsymbol{\delta}^*}(\boldsymbol{x}_p^{(i)})\|_\infty = \|\frac{1}{|G|}\sum_{i\in G} \nabla_{\boldsymbol{\delta}}\hat{z}_{\boldsymbol{\delta}^*}(\boldsymbol{x}_p^{(i)})\|_\infty,
$$

we only need to bound $\left\| \frac{1}{|G|} \sum_{i \in G} \nabla_{\boldsymbol{\delta}} \hat{z}_{\boldsymbol{\delta}^*}(\boldsymbol{x}_p^{(i)}) \right\|_\infty$. As samples in $G$ are i.i.d. samples drawn from $P$, here can we invoke the Lemma 2 from [14]. First we need the following conditions:

**Assumption 6.** *For any vector $\boldsymbol{u} \in \mathbb{R}^{dim(\boldsymbol{\delta}^*)}$ such that $\boldsymbol{\delta}^* + \boldsymbol{u} \in \mathrm{Ball}(\rho)$, the Hessian of the likelihood function, $\nabla^2 \mathcal{L}(\boldsymbol{\delta}^* + \boldsymbol{u})$, has a bounded spectral norm, i.e., $\|\nabla^2 \mathcal{L}(\boldsymbol{\delta}^* + \boldsymbol{u})\| \leq \lambda_{\max}$.*

**Assumption 7** (Smooth Density Ratio Model Assumption). *For any vector $\boldsymbol{u} \in \mathbb{R}^{dim(\boldsymbol{\delta}^*)}$ such that $\boldsymbol{\delta}^* + \boldsymbol{u} \in \mathrm{Ball}(\rho)$ and every $a \in \mathbb{R}$, the following inequality holds:*

$$\mathbb{E}_q \left[ \exp \left( a \left( r(\boldsymbol{x}, \boldsymbol{\delta}^* + \boldsymbol{u}) - 1 \right) \right) \right] \leq \exp \left( K a^2 \right).$$

If $n_q = \Omega(|G|^2)$, and $\lambda_n \geq \sqrt{\frac{K_1 \log d}{|G|}}$, according to Lemma 2 from [14] we have

$$P \left( \left\| \frac{1}{|G|} \sum_{i \in G} \nabla_{\boldsymbol{\delta}} \hat{z}_{\boldsymbol{\delta}^*}(\boldsymbol{x}_p^{(i)}) \right\|_\infty \geq \lambda_n \right) \leq \exp \left( -c_1 |G| \right), \tag{29}$$

where $K_1$ and $c_1$ are constants. Finally, we can re-state the Theorem 1 using $\kappa_2 = 0$, $\tau_2 = 0$ and (29) to obtain Theorem 2.

## 8.6  Proof of Theorem 3

First we verify (11).

**Lemma 3.** *Under Assumptions 4 and 5,*

$$\|\nabla_{\boldsymbol{\delta}} \mathcal{L}(\hat{\boldsymbol{\delta}}, \hat{\boldsymbol{w}}) - \nabla_{\boldsymbol{\delta}} \mathcal{L}(\hat{\boldsymbol{\delta}}, \boldsymbol{w}^*)\|_\infty \leq 2 C_{\mathrm{CDF}} \cdot \|\boldsymbol{u}\| C_p + \frac{2L \cdot C_p}{\sqrt{n_p}},$$

*where $L$ is a positive constant. The second term reflects the cost of using the empirical sample to control the $\nu$-th quantile in (28).*

Therefore

$$\langle \nabla_{\boldsymbol{\delta}} \mathcal{L}(\hat{\boldsymbol{\delta}}, \hat{\boldsymbol{w}}) - \nabla_{\boldsymbol{\delta}} \mathcal{L}(\hat{\boldsymbol{\delta}}, \boldsymbol{w}^*), \boldsymbol{u} \rangle \geq - \left( 2 C_{\mathrm{CDF}} \cdot \|\boldsymbol{u}\| C_p + \frac{2L \cdot C_p}{\sqrt{n_p}} \right) \|\boldsymbol{u}\|_1^2$$

$$\geq - 2\sqrt{k} C_{\mathrm{CDF}} C_p \|\boldsymbol{u}\|^2 - \frac{2L \cdot C_p \|\boldsymbol{u}\|_1}{\sqrt{n_p}}.$$

It can be seen that $\kappa_2 = 2\sqrt{k} C_{\mathrm{CDF}} C_p$, $\tau_2(n, d) = \frac{2L \cdot C_p}{\sqrt{n_p}}$. The proof of Lemma 3 uses a fact that only $\boldsymbol{x}_p$ in the "zone" $T(\boldsymbol{u}, \frac{L_1}{\sqrt{n_p}})$ are "dangerous" as they may be mistakenly included or missed out under small perturbation of $\boldsymbol{u}$. See Section 8.8 in Appendix for the proof.

To show $\|\nabla_{\boldsymbol{\delta}} \mathcal{L}(\boldsymbol{\delta}^*, \boldsymbol{w}^*)\|_\infty \to 0$, we need some extra procedures since $z_{\boldsymbol{\delta}^*}(\boldsymbol{x}_q)$ are not necessarily upper-bounded by $t(\boldsymbol{\delta}^*)$. The following lemma bounds $\|\nabla_{\boldsymbol{\delta}} \mathcal{L}(\boldsymbol{\delta}^*, \boldsymbol{w}^*)\|_\infty$.

**Lemma 4.** *Under Assumptions 1, 5, 6 and 7 holds, and if*

$$\lambda_n \geq \sqrt{\frac{K_1' \log d}{|\overline{X}^p(\boldsymbol{\delta}^*)|}} + \frac{2 C_r^2 C_q |X_q \backslash \overline{X}^q(\boldsymbol{\delta}^*)|}{n_q} \tag{30}$$

$\|\nabla_{\boldsymbol{\delta}} \mathcal{L}(\boldsymbol{\delta}^*, \boldsymbol{w}^*)\|_\infty \leq \lambda_n$ *with probability at least $1 - \exp(c_1' |\overline{X}^p(\boldsymbol{\delta}^*)|)$, where $c_1'$ and $K_1'$ are constants,*

See Section 8.7 in Appendix for the proof.

Finally, we can restate Theorem 1 as Theorem 3 using $\kappa_1 = \frac{\nu \kappa_1'}{2 C_r^2}$, $\tau_1(n, d) = \frac{\nu c}{2 C_r^2 \sqrt{n_q}}$, $\kappa_2 = 2\sqrt{k} C_{\mathrm{CDF}} C_p$, $\tau_2(n, d) = \frac{2L \cdot C_p}{\sqrt{n_p}}$ and (30), making sure that $\kappa_1 > \kappa_2$.

Figure 4: An illustration of $B$ and $G$ in the case of truncation setting. In this setting, we treat $X_q \backslash \overline{X}^q(\boldsymbol{\delta}^*)$ as a kind of outlier of $Q$ and only appear in very small quantity.

## 8.7 Proof of Lemma 4

First, we recycle some notations from the previous section: $G := \overline{X}^q(\boldsymbol{\delta}^*), B := X_q \backslash \overline{X}^q(\boldsymbol{\delta}^*)$. The reason for this arrangement can be seen from Figure 4.

Denote $e^{(j)} := \exp\left[\langle \boldsymbol{\delta}^*, \boldsymbol{x}_q^{(j)} \rangle\right], s := \sum_{j=1}^{n_q} e^{(j)}$ and $\bar{s} = \sum_{i \in G} e^{(i)}$. Note that

$$\nabla_{\boldsymbol{\delta}} \mathcal{L}(\boldsymbol{\delta}^*, \boldsymbol{w}^*) = \frac{1}{n_p} \sum_{i \in \overline{X}^p(\boldsymbol{\delta}^*)} \left[ \boldsymbol{x}_p^{(i)} - \nabla_{\boldsymbol{\delta}} \log \widehat{N}(\boldsymbol{\delta}^*) \right].$$

$$\|\nabla_{\boldsymbol{\delta}} \mathcal{L}(\boldsymbol{\delta}^*, \boldsymbol{w}^*)\|_\infty = \| \frac{1}{n_p} \sum_{i \in \overline{X}^p(\boldsymbol{\delta}^*)} \left[ \boldsymbol{x}_p^{(i)} - \nabla_{\boldsymbol{\delta}} \log \widehat{N}(\boldsymbol{\delta}^*) \right] \|_\infty$$

$$= \frac{1}{n_p} \| \sum_{i \in \overline{X}^p(\boldsymbol{\delta}^*)} \left[ \boldsymbol{x}_p^{(i)} - \sum_{j=1}^{n_q} \frac{e^{(j)}}{s} \boldsymbol{x}_q^{(j)} \right] \|_\infty$$

$$= \frac{1}{n_p} \| \sum_{i \in \overline{X}^p(\boldsymbol{\delta}^*)} \left[ \boldsymbol{x}_p^{(i)} - \sum_{j \in G} \frac{e^{(j)}}{s} \boldsymbol{x}_q^{(j)} - \sum_{j \in B} \frac{e^{(j)}}{s} \boldsymbol{x}_q^{(j)} \right] \|_\infty$$

$$= \frac{1}{n_p} \| \sum_{i \in \overline{X}^p(\boldsymbol{\delta}^*)} \left[ \boldsymbol{x}_p^{(i)} - \frac{\bar{s}}{s} \sum_{j \in G} \frac{e^{(j)}}{\bar{s}} \boldsymbol{x}_q^{(j)} - \sum_{j \in B} \frac{e^{(j)}}{s} \boldsymbol{x}_q^{(j)} \right] \|_\infty$$

$$= \frac{1}{n_p} \| \sum_{i \in \overline{X}^p(\boldsymbol{\delta}^*)} \left[ \boldsymbol{x}_p^{(i)} - \sum_{j \in G} \frac{e^{(j)}}{\bar{s}} \boldsymbol{x}_q^{(j)} + (1 - \frac{\bar{s}}{s}) \sum_{j \in G} \frac{e^{(j)}}{\bar{s}} \boldsymbol{x}_q^{(j)} - \sum_{j \in B} \frac{e^{(j)}}{s} \boldsymbol{x}_q^{(j)} \right] \|_\infty$$

$$\leq \underbrace{\frac{1}{n_p} \| \sum_{i \in \overline{X}^p(\boldsymbol{\delta}^*)} \boldsymbol{x}_p^{(i)} - \sum_{j \in G} \frac{e^{(j)}}{\bar{s}} \boldsymbol{x}_q^{(j)} \|_\infty}_{a(n,d)} + \|(1 - \frac{\bar{s}}{s}) \sum_{j \in G} \frac{e^{(j)}}{\bar{s}} \boldsymbol{x}_q^{(j)} - \sum_{j \in B} \frac{e^{(j)}}{s} \boldsymbol{x}_q^{(j)} \|_\infty$$

$$\leq a(n,d) + \frac{s - \bar{s}}{s} \sum_{j \in G} \| \frac{e^{(j)}}{\bar{s}} \boldsymbol{x}_q^{(j)} \|_\infty + \sum_{j \in B} \| \frac{e^{(j)}}{s} \boldsymbol{x}_q^{(j)} \|_\infty$$

$$\leq a(n,d) + \frac{C_r^2 |B|}{n_q} \cdot \frac{1}{|G|} \sum_{j \in G} \| \boldsymbol{x}^{(j)} \|_\infty + \frac{C_r}{n_q} \sum_{j \in B} \| \boldsymbol{x}_q^{(j)} \|_\infty$$

$$\leq a(n,d) + \frac{C_r^2 |B| C_q}{n_q} + \frac{C_r |B| C_q}{n_q}$$

$$\leq a(n,d) + \frac{2 C_r^2 |B| C_q}{n_q}$$

Now, as $\overline{X}^p(\boldsymbol{\delta}^*)$ and $G$ contains only i.i.d. samples and due to the definition of $\boldsymbol{\delta}^*$, we can invoke Lemma 2 again from [14] to bound $a(n,d)$. That is if Assumptions 6 and 7 hold and $n_q = \Omega(n_p^2)$, and $\lambda_n \geq \sqrt{\frac{K_1' \log d}{|\overline{X}^p(\boldsymbol{\delta}^*)|}}$

$$P\left(a(n,d) \geq \lambda_n\right) \leq \exp\left(-c_1' |\overline{X}^p(\boldsymbol{\delta}^*)|\right), \tag{31}$$

where $K_1'$ and $c_1'$ are constants. By taking the extra $\frac{2C_r^2 |B| C_q}{n_q}$ into account, we obtain Lemma 4.

## 8.8 Proof of Lemma 3

Before we start, we need to define a few empirical counterparts of population quantities used in Section 5.3.

- $P_n$ is the empirical distribution of $P$.
- $\hat{t}(\boldsymbol{\delta})$ is the empirical version of $t(\boldsymbol{\delta})$ and is defined according to

$$P_{n_p}\left[\hat{z}_{\boldsymbol{\delta}} < \hat{t}_\nu(\boldsymbol{\delta}))|X_q\right] \leq \nu, \quad P_{n_p}\left[\hat{z}_{\boldsymbol{\delta}} \leq \hat{t}_\nu(\boldsymbol{\delta}))|X_q\right] \geq \nu$$

- The set $\overline{X}_n(\boldsymbol{\delta})$ is similar to $\overline{X}(\boldsymbol{\delta})$ but defined by $\hat{z}$ and $\hat{t}$:

$$\overline{X}_n(\boldsymbol{\delta}) := \left\{\boldsymbol{x} \in \mathbb{R}^d | \hat{z}_{\boldsymbol{\delta}}(\boldsymbol{x}) < \hat{t}(\boldsymbol{\delta})\right\}.$$

- $\overline{X}_n^p(\boldsymbol{\delta}) := X_p \cap \overline{X}_n(\boldsymbol{\delta})$.
- The "borderline points" of $X_p$: $X_{\text{border}}(\boldsymbol{\delta}) := \{\boldsymbol{x} \in X_p | \hat{z}_{\boldsymbol{\delta}}(\boldsymbol{x}) = \hat{t}_\nu(\boldsymbol{\delta}))\}$.

*Proof.* We first expand $\|\nabla_{\boldsymbol{\delta}}\mathcal{L}(\hat{\boldsymbol{\delta}}, \hat{\boldsymbol{w}}) - \nabla_{\boldsymbol{\delta}}\mathcal{L}(\hat{\boldsymbol{\delta}}, \boldsymbol{w}^*)\|_\infty$ as

$$\|\nabla_{\boldsymbol{\delta}}\mathcal{L}(\boldsymbol{\delta}^* + \hat{\boldsymbol{u}}, \hat{\boldsymbol{w}}) - \nabla_{\boldsymbol{\delta}}\mathcal{L}(\boldsymbol{\delta}^* + \hat{\boldsymbol{u}}, \boldsymbol{w}^*)\|_\infty$$

$$= \| \sum_{i, w_i \neq 0} \hat{w}_i \boldsymbol{x}_p^{(i)} - \frac{1}{n_p} \sum_{i \in \overline{X}^p(\boldsymbol{\delta}^*)} \boldsymbol{x}_p^{(i)} \|_\infty$$

$$\leq \frac{1}{n_p} \sum_{i \in \underbrace{\overline{X}_n^p(\boldsymbol{\delta}^* + \hat{\boldsymbol{u}}) \backslash \overline{X}^p(\boldsymbol{\delta}^*)}_{M_1(\hat{\boldsymbol{u}})}} \|\boldsymbol{x}_p^{(i)}\|_\infty + \frac{1}{n_p} \sum_{i \in \underbrace{\overline{X}^p(\boldsymbol{\delta}^*) \backslash \overline{X}_n^p(\boldsymbol{\delta}^* + \hat{\boldsymbol{u}})}_{M_2(\hat{\boldsymbol{u}})}} \|\boldsymbol{x}_p^{(i)}\|_\infty$$

$$+ \frac{1}{n_p} \sum_{i \in X_{\text{border}}(\boldsymbol{\delta}^* + \hat{\boldsymbol{u}})} \|\boldsymbol{x}_p^{(i)}\|_\infty$$

$$= \frac{1}{n_p} \sum_{i \in M(\hat{\boldsymbol{u}})} \|\boldsymbol{x}_p^{(i)}\|_\infty + \frac{1}{n_p} \sum_{i \in X_{\text{border}}(\boldsymbol{\delta}^* + \hat{\boldsymbol{u}})} \|\boldsymbol{x}_p^{(i)}\|_\infty, \tag{32}$$

where $M(\boldsymbol{u}) := M_1(\boldsymbol{u}) \cup M_2(\boldsymbol{u})$, given $\boldsymbol{u} \in \text{Ball}(\rho)$. Note we isolate the borderline points $X_{\text{border}}$ in our analysis as they may have interior weights, i.e., $w_i \in [0, \frac{1}{n_p}]$.

We first figure out the cardinality of $M(\boldsymbol{u})$, a set where samples are *likely* to be "misplaced" to the other set under a small perturbation. However, direct quantifying $M(\boldsymbol{u})$ is hard but we now show that $M(\boldsymbol{u}) \subseteq X_p \cap T(\boldsymbol{u}, \epsilon)$ whose cardinality is bounded by our assumptions. See Figure 5 for details.

First, we show that if $z_{\boldsymbol{\delta}^*}(\boldsymbol{x}_p) \geq t(\boldsymbol{\delta}^*) + 2C_{\text{lip}}\|\boldsymbol{u}\| + \epsilon$, then $\boldsymbol{x}_p \notin \overline{X}^p(\boldsymbol{\delta}^*) \cup \overline{X}_n^p(\boldsymbol{\delta}^* + \boldsymbol{u})$. As we will see, $\epsilon \in (0,1)$ is chosen afterwards.

Under this setting, obviously, $\boldsymbol{x}_p \notin \overline{X}^p(\boldsymbol{\delta}^*)$, thus it is suffice to show that $\boldsymbol{x}_p \notin \overline{X}_n^p(\boldsymbol{\delta}^* + \boldsymbol{u})$. Note that for any constant c, the quantile of $z' := z(\boldsymbol{\delta}^*) + c$ is $t(\boldsymbol{\delta}^*) + c$.

Since $z_{\boldsymbol{\delta}^*}$ and $\hat{z}_{\boldsymbol{\delta}^*}$ differ only by their normalization functions, we have $z_{\boldsymbol{\delta}^*}(\boldsymbol{x}_p) - t(\boldsymbol{\delta}^*) = \hat{z}_{\boldsymbol{\delta}^*}(\boldsymbol{x}_p) - t'(\boldsymbol{\delta}^*)$, where $t'(\boldsymbol{\delta}^*)$ is defined as $P\left[\hat{z}_{\boldsymbol{\delta}^*} < t'_\nu(\boldsymbol{\delta}^*))|X_q\right] \leq \nu$ and $P\left[\hat{z}_{\boldsymbol{\delta}^*} \leq t'_\nu(\boldsymbol{\delta}^*))|X_q\right] \geq \nu$ for a given $X_q$, so we have $\hat{z}_{\boldsymbol{\delta}^*}(\boldsymbol{x}_p) \geq t'(\boldsymbol{\delta}^*) + 2C_{\text{lip}}\|\boldsymbol{u}\| + \epsilon$. Combining this inequality with Assumption 4, we have

$$\hat{z}_{\boldsymbol{\delta}^* + \boldsymbol{u}}(\boldsymbol{x}_p) \geq \hat{z}_{\boldsymbol{\delta}^*}(\boldsymbol{x}_p) - C_{\text{lip}}\|\boldsymbol{u}\| \geq t'(\boldsymbol{\delta}^*) + C_{\text{lip}}\|\boldsymbol{u}\| + \epsilon \tag{33}$$

Figure 5: The relationship of $\overline{X}^P(\boldsymbol{\delta}^*)$, $\overline{X}_n^P(\boldsymbol{\delta}^* + \boldsymbol{u})$, $X_p \cap T(\boldsymbol{u}, \epsilon)$ and $X_{\text{border}}(\boldsymbol{\delta}^* + \boldsymbol{u})$.

From Dvoretzky–Kiefer–Wolfowitz inequality if $n_p$ is large enough, with high probability $\left| t'(\boldsymbol{\delta}^*) - \hat{t}(\boldsymbol{\delta}^*) \right| \leq \frac{L_1}{\sqrt{n_p}} \leq 1$ which is independent of the choice of $X_q$. Thus we set $\epsilon = \frac{L_1}{\sqrt{n_p}}$, and

$$t'(\boldsymbol{\delta}^*) + \frac{L_1}{\sqrt{n_p}} + C_{\text{lip}} ||\boldsymbol{u}|| \geq \hat{t}(\boldsymbol{\delta}^*) + C_{\text{lip}} ||\boldsymbol{u}|| \quad \text{w.h.p.} \tag{34}$$

From Assumption 4, $\hat{z}_{\boldsymbol{\delta}^* + \boldsymbol{u}}$ and $\hat{z}_{\boldsymbol{\delta}^*}$ differ only by $C_{\text{lip}} ||\boldsymbol{u}||$, which means their $\nu$-percentile $\hat{t}(\boldsymbol{\delta}^* + \boldsymbol{u})$ and $\hat{t}(\boldsymbol{\delta}^*)$ differ by $C_{\text{lip}} ||\boldsymbol{u}||$ at most. Thus,

$$\hat{t}(\boldsymbol{\delta}^*) + C_{\text{lip}} ||\boldsymbol{u}|| \geq \hat{t}(\boldsymbol{\delta}^* + \boldsymbol{u}) \tag{35}$$

From (33) (34) and (35), we now have $\hat{z}_{\boldsymbol{\delta}^* + \boldsymbol{u}}(\boldsymbol{x}_p) \geq \hat{t}(\boldsymbol{\delta}^* + \boldsymbol{u})$ which means

$$\boldsymbol{x}_p \notin \overline{X}_n^p(\boldsymbol{\delta}^* + \boldsymbol{u})$$

with high probability. As we have mentioned earlier, it is obvious that $\boldsymbol{x}_p \notin \overline{X}^P(\boldsymbol{\delta}^*)$, so

$$\boldsymbol{x}_p \notin \overline{X}^P(\boldsymbol{\delta}^*) \cup \overline{X}_n^p(\boldsymbol{\delta}^* + \boldsymbol{u}). \tag{36}$$

Similarly, one can show if $z_{\boldsymbol{\delta}^*}(\boldsymbol{x}_p) \leq t(\boldsymbol{\delta}^*) - 2C_{\text{lip}} ||\boldsymbol{u}|| - \epsilon$, then

$$\boldsymbol{x}_p \in \overline{X}_n^p(\boldsymbol{\delta}^* + \boldsymbol{u}) \cap \overline{X}^P(\boldsymbol{\delta}^*) \tag{37}$$

(which is the center-most region in Figure 5) with high probability. Now we can conclude that:

$$M(\boldsymbol{u}) \subseteq X_p \cap T(\boldsymbol{u}, \frac{L_1}{\sqrt{n_p}}) \quad \text{w.h.p.} \tag{38}$$

Due to Dvoretzky–Kiefer–Wolfowitz inequality,

$$P_{n_p}(\boldsymbol{x}_p \in T(\boldsymbol{u}, \frac{L_1}{\sqrt{n_p}})) - P(\boldsymbol{x}_p \in T(\boldsymbol{u}, \frac{L_1}{\sqrt{n_p}})) \leq \frac{L_2}{\sqrt{n_p}}$$

holds with probability at least $\exp\left[ -2L_2^2 \right], \forall L_2 > 0$. Thus, using Assumption 4 we have

$$P_{n_p}(\boldsymbol{x}_p \in T(\boldsymbol{u}, \frac{L_1}{\sqrt{n_p}})) \leq C_{\text{CDF}} \cdot ||\boldsymbol{u}|| + \frac{L_1}{\sqrt{n_p}} + \frac{L_2}{\sqrt{n_p}} \quad \text{w.h.p.}$$

Now we know the cardinality of $X_p \cap T(\boldsymbol{u}, \frac{L_1}{\sqrt{n_p}})$ can be bounded by $\left( C_{\text{CDF}} \cdot ||\boldsymbol{u}|| + \frac{L_1 + L_2}{\sqrt{n_p}} \right) \cdot n_p$ with high probability. Finally, we have

$$\frac{1}{n_p} \sum_{i \in X_p \cap T(\boldsymbol{u}, \frac{L_1}{\sqrt{n_p}})} ||\boldsymbol{x}_p^{(i)}||_\infty \leq \frac{1}{n_p} \left( C_{\text{CDF}} \cdot ||\boldsymbol{u}|| + \frac{L_1 + L_2}{\sqrt{n_p}} \right) \cdot n_p C_p$$

$$\leq C_{\text{CDF}} \cdot ||\boldsymbol{u}|| C_p + \frac{(L_1 + L_2) \cdot C_p}{\sqrt{n_p}} \tag{39}$$

Now, we show $X_{\text{border}}(\boldsymbol{\delta}^* + \boldsymbol{u}) \subseteq X_p \cap T(\boldsymbol{u}, \frac{L_1}{\sqrt{n_p}})$. The proof for this is similar to the arguments above. Using Assumption 4, it can be shown that

$$\hat{t}(\boldsymbol{\delta}^* + \boldsymbol{u}) \in \left[ t'(\boldsymbol{\delta}^*) - C_{\text{lip}}\|\boldsymbol{u}\| - \frac{L_1}{\sqrt{n_p}}, t'(\boldsymbol{\delta}^*) + C_{\text{lip}}\|\boldsymbol{u}\| + \frac{L_1}{\sqrt{n_p}} \right],$$

and from definition, $\forall \boldsymbol{x} \in X_{\text{border}}(\boldsymbol{\delta}^* + \boldsymbol{u}), \hat{z}_{\boldsymbol{\delta}^*+\boldsymbol{u}}(\boldsymbol{x}) = \hat{t}(\boldsymbol{\delta}^* + \boldsymbol{u})$,

$$\hat{z}_{\boldsymbol{\delta}^*+\boldsymbol{u}}(\boldsymbol{x}) \in \left[ t'(\boldsymbol{\delta}^*) - C_{\text{lip}}\|\boldsymbol{u}\| - \frac{L_1}{\sqrt{n_p}}, t'(\boldsymbol{\delta}^*) + C_{\text{lip}}\|\boldsymbol{u}\| + \frac{L_1}{\sqrt{n_p}} \right],$$

and due to Assumption 4,

$$\hat{z}_{\boldsymbol{\delta}^*}(\boldsymbol{x}) \in \left[ t'(\boldsymbol{\delta}^*) - 2C_{\text{lip}}\|\boldsymbol{u}\| - \frac{L_1}{\sqrt{n_p}}, t'(\boldsymbol{\delta}^*) + 2C_{\text{lip}}\|\boldsymbol{u}\| + \frac{L_1}{\sqrt{n_p}} \right].$$

Again, this relationship does not change if we replace $\hat{z}$ and $t'$ at the same time with $z$ and $t$

$$z_{\boldsymbol{\delta}^*}(\boldsymbol{x}) \in \left[ t(\boldsymbol{\delta}^*) - 2C_{\text{lip}}\|\boldsymbol{u}\| - \frac{L_1}{\sqrt{n_p}}, t(\boldsymbol{\delta}^*) + 2C_{\text{lip}}\|\boldsymbol{u}\| + \frac{L_1}{\sqrt{n_p}} \right] \subseteq T(\boldsymbol{u}, \frac{L_1}{\sqrt{n_p}}). \quad (40)$$

Inequalities (32), (38), (39) and (40) complete the proof. □

# 9 Numerical Analysis

In this section, we present a few numerical experimental results under outlier and truncation setting. In all experiments, we set $n_p = n_q = 5000, \lambda = 0$, and the solution of $\hat{\boldsymbol{\delta}}$ was obtained using Algorithm 1. We let $f(x) = x$. Note this is the correct log-ratio model for two Gaussian distributions with different means.

**Outlier Setting**    In this setting, we first generate two "good" datasets $G \overset{\text{i.i.d.}}{\sim} p(x) = N(0, 1)$, and $X_q \overset{\text{i.i.d.}}{\sim} q(x) = N(-.75, 1)$. The outlier set $B_b$ is generated from a uniform distribution $U(-0.4 + b, 0.4 + b), b \in [0, 6]$. The density ratio estimation is performed using two sets of data: $X_{p,b} = \{G, B_b\}$ and $X_q$, where the cardinality of $B$ is 1000. We repeat the estimation using different choices of $b$ and test its influence on our estimate $\hat{r}(\boldsymbol{x}; \hat{\boldsymbol{\delta}}_b)$. The results can be seen from Figure 6, where the histograms of $G$ and $X_q$ are colored red and green respectively. The true density ratio $\frac{p(x)}{q(x)}$ is plotted as a dotted line. The histograms of $B_b$ with different choices of $b$ was plotted using gradient colors from light blue to purple (we skipped some choices of $b$ for better visualization). For each $b$, we run the density ratio estimation, and plot learned $\hat{r}(\boldsymbol{x}; \hat{\boldsymbol{\delta}}_b)$ using the same gradient color. In the figure, we resale $\hat{r}(\boldsymbol{x}; \hat{\boldsymbol{\delta}}_b)$ and the true density ratio using a same constant, so they can be plotted alongside with the histogram. Here, we test two methods: the log-Linear KLIEP and the robust estimator proposed in this paper.

It can be easily seen that as $b \to 6$, KLIEP (Figure 6a) tends to significantly overestimate the density ratio and is sensitive to the change of $b$. The proposed method (Figure 6b), tends to underestimate the density ratio when $b$ is small. However, as $b$ gradually shifts away from the center of $X_p$, leaving the "gap" between inlier and outlier, the robust estimator converges to the true density ratio function.

**Truncated Setting**    In this setting, we generate samples $X_p \overset{\text{i.i.d.}}{\sim} p(x) = N(0, 1)$ without any contamination. Usually, the $\nu$-th quantile of $z(\boldsymbol{x}_p; \boldsymbol{\delta}^*)$ cannot be analytically computed as we do not know the true density ratio. However, it can be seen that for a strictly monotone increasing $z(\boldsymbol{x}_p, \boldsymbol{\delta}^*)$, samples in the $\nu$-th quantile of $z(\boldsymbol{x}_p, \boldsymbol{\delta}^*)$ must be in the $\nu$-th quantile of $x_p$ since the relative order among $x_p$ is preserved after a strictly monotone transform. Thus, we obtain the truncation domain $\overline{X}(\boldsymbol{\delta}^*) = \{-\infty \le x \le \Phi^{-1}(\nu)\}$, where $\Phi^{-1}$ is the inverse CDF of $N(0, 1)$. We then generate samples $X_q \sim TN(-0.5, 1, -\infty, \Phi^{-1}(\nu))$, where $TN$ is a truncated Gaussian distribution and the last two parameters are the truncation borders. Note we set the mean of $Q$ to be a negative value so that the true density ratio $\bar{p}/\bar{q}$ is a monotone increasing function.

The results for $\nu = 0.5$ are plotted on Figure 7 where the true truncated ratio is plotted as a dotted line. It can be seen that the learned $\hat{r}(\boldsymbol{x}; \hat{\boldsymbol{\delta}})$ is fairly close to the true truncated density ratio.

(a) Non-robust Density Ratio Estimation
(b) Robust Density Ratio Estimation

Figure 6: Outlier Setting

Figure 7: Truncated Setting

## Footnotes

[5]if $t = 0$ is the optimal and assume $R(\mathbf{0}) = 0$, we only have a trivial solution $\boldsymbol{\delta} = \mathbf{0}, \boldsymbol{\epsilon} = \mathbf{0}$, which is easy to verify and rules out.