[Reviews · NeurIPS 2017]

Reviewer 1



This paper studies a robust approach for density ratio estimation, which is an important problem. The paper is well-written. The paper establishes consistency under different settings. The authors further provide interesting numerical studies to backup their methods. My only question is: under the high-dimensional setting, what scaling can the method tolerate? The authors should explicitly list this scaling.

Reviewer 2



Summary: This paper proposes a "trimmed" estimator that robustly (to outliers) estimates the ratio of two densities, assuming an a exponential family model. This robustness is important, as density ratios can inherently be very unstable when the denominator is small. The proposed model is based on an optimization problem, motivated by minimizing KL divergence between the two densities in the ratio, and is made more computationally tractable by re-expressing it in terms of an equivalent saddle-point/max-min formulation. Similar to the one-class SVM, this formulation explicitly discards a portion (determined by a tuning parameter) of "outlier" samples. The density-ratio estimator is shown to be consistent in two practical settings, one in which the data contains a small portion of explicit outliers and another in which the estimand is intrinsically unstable. Finally, experiments are presented in the context of change detection, including a synthetic Gaussian example and a real relative object detection example. Main Comments: The paper is fairly clearly written, and the derivation of the estimator is well motivated throughout. While the application is somewhat specific, the method seems practically and theoretically strong, and I appreciate that the theoretical results are tailored to the application settings of interest. The experients aren't a particularly thorough comparison, but nevertheless effectively illustrate the behavior and benefits of the proposed estimator. Minor Comments: Line 22: "a few sample" should be "a few samples" Line 29: small typo: "uses the log-linear models" should be "uses the log-linear model" or "uses log-linear models" The placement of Section 3.2 is a bit odd, as it interupts the flow of deriving the estimator. Perhaps this could go immediately before or after the experimental results? Line 131: "The key of" should be "The key to" Figure 3: Perhaps include a note saying that this figure is best viewed in color, as I found this figure very confusing in my greyscale printout.

Reviewer 3



Trimmed Density Ratio Estimation This paper proposes a convex method for robust density ratio estimation, and provides several theoretical and empirical results. I found the method to be well motivated, and of course Figure 3 is quite fun. Based on the evidence in the paper, I would feel comfortable recommending the method to others. I would be happy to see these results published; however, I found the presentation to be fairly terse, and thought the paper was very difficult to follow. My main recommendation for the writing would be to focus more on core contributions, and to remove some side remarks to allow for longer discussions. Moreover, the paper was missing some references. Noting these issues, my current recommendation for the paper is a middle-of-the-road “accept”; however, if the authors are willing to thoroughly revise their paper, I would be open to considering a higher rating. Below are some specific comments. Comments on motivation + literature: - The paper opens the abstract by describing density ratio estimation (DRE) as a “versatile tool” in machine learning. I think this substantially undersells the problem: DRE is a a core statistical and machine learning task, and any advance to it should be of broad interest. I would recommend stressing the many applications of DRE more. - On a related note, DRE has also received quite a bit of attention in the statistics literature. For example, Efron and Tibshirani (1996) use it to calibrate kernel estimators, and Fithian and Wager (2016) use DRE to stabilize heavy-tailed AB-tests. Fokianos (2004) models the density ratio to fit several densities at once. This literature should also be discussed. - Are there some applications where trimming is better justified than others? Engineering applications relating to GANs or anomaly detection seem like very natural places to apply trimming; other problems, like two-sample testing, may be more of a stretch. (In applications like anomaly detection, it’s expected to have regions where the density ratio diverges, so regularizing against them is natural. On the other hand, if one finds a diverging density ratio in two-sample testing, that’s very interesting and shouldn’t be muted out.) Comments on relationship to the one-class SVM: - I found Section 2 on the one-class SVM to be very distracting. It’s too short to help anyone who isn’t already familiar with the method, and badly cuts the flow of the paper. I think a version of the paper that simply removed Section 2 would be much nicer to read. (The discussion of the one-class SVM is interesting, but belongs in the discussion section of a long journal paper, not on page 2 of an 8-page NIPS paper.) - The use of the one-SVM in the experiments was interesting. However, I think the median reader would benefit from a lot more discussion. The point is that the proposed method gets to leverage the “background” images, whereas one-SVM doesn’t. So even though one-SVM is a good method, you bring more data to the table, so there’s no way one-SVM could win. (I know that the current draft says this, but it says it so briefly that I doubt anyone would notice it if they were reading at a regular speed.) - As a concrete recommendation: If the authors were to omit Section 2 and devote the resulting space to adding a paragraph or two to the intro that reviews and motivates the DRE literature to the intro (including the stats literature) and to thoroughly contrasting their results with the one-SVM in the results section, I think the paper would be much easier to read. Other comments: - In the setting of 3.1, Fithian and Wager (2016) show that if we assume that one class has much more data than the other, then density ratio models can be fit via logistic regression. Can the authors comment on how the proposed type of trimming can be interpreted in the logistic regression case? - Section 3.2 is awkwardly placed, and cuts the flow. I recommend moving this material to the intro. - I wasn’t sold by the connection to the SVM on lines 120-129. The connection seems rather heuristic to me (maybe even cosmetic?); I personally would omit it. The core contributions of the paper are strong enough that there’s no need to fish for tentative connections. - The most classical approach to convex robust estimation is via Huberization (e.g., Huber, 1972). What’s recommended here seems very different to me —- I think readers may benefit from a discussion of how they’re different. - The approach of assuming a high-dimensional linear model for the theory and and RBF kernel for the experiments is fine. Highlighting this choice more prominently might help some readers, though. References: Efron, Bradley, and Robert Tibshirani. "Using specially designed exponential families for density estimation." The Annals of Statistics 24.6 (1996): 2431-2461. Fithian, William, and Stefan Wager. "Semiparametric exponential families for heavy-tailed data." Biometrika 102.2 (2014): 486-493. Fokianos, Konstantinos. "Merging information for semiparametric density estimation." Journal of the Royal Statistical Society: Series B (Statistical Methodology) 66.4 (2004): 941-958. Huber, Peter J. “Robust statistics: A review." The Annals of Mathematical Statistics (1972): 1041-1067. ########## UPDATE In response to the authors’ comprehensive reply, I have increased my rating to an “8”.